# Reproducible brain-wide association studies require thousands of individuals

Scott Marek[1,30 ✉], Brenden Tervo-Clemmens[2,3,30 ✉], Finnegan J. Calabro[4,5], David F. Montez[6], Benjamin P. Kay[6], Alexander S. Hatoum[1], Meghan Rose Donohue[1], William Foran[4], Ryland L. Miller[1,6], Timothy J. Hendrickson[7], Stephen M. Malone[8], Sridhar Kandala[1], Eric Feczko[9,10], Oscar Miranda-Dominguez[9,10], Alice M. Graham[11], Eric A. Earl[9,11], Anders J. Perrone[9,11], Michaela Cordova[11], Olivia Doyle[11], Lucille A. Moore[11], Gregory M. Conan[9,11], Johnny Uriarte[11], Kathy Snider[11], Benjamin J. Lynch[9,12], James C. Wilgenbusch[9,12], Thomas Pengo[7], Angela Tam[13,14,15,16], Jianzhong Chen[13,14,15,16], Dillan J. Newbold[6], Annie Zheng[6], Nicole A. Seider[6], Andrew N. Van[6,17], Athanasia Metoki[6], Roselyne J. Chauvin[6], Timothy O. Laumann[1], Deanna J. Greene[18], Steven E. Petersen[6,17,19,20,21], Hugh Garavan[22], Wesley K. Thompson[23], Thomas E. Nichols[24], B. T. Thomas Yeo[13,14,15,16,25,26], Deanna M. Barch[1,21], Beatriz Luna[3,4], Damien A. Fair[9,10,27,31 ✉] & Nico U. F. Dosenbach[6,17,19,28,29,31 ✉]

Magnetic resonance imaging (MRI) has transformed our understanding of the human brain through well-replicated mapping of abilities to specific structures (for example, lesion studies) and functions[1–3] (for example, task functional MRI (fMRI)). Mental health research and care have yet to realize similar advances from MRI. A primary challenge has been replicating associations between inter-individual differences in brain structure or function and complex cognitive or mental health phenotypes (brain-wide association studies (BWAS)). Such BWAS have typically relied on sample sizes appropriate for classical brain mapping[4] (the median neuroimaging study sample size is about 25), but potentially too small for capturing reproducible brain–behavioural phenotype associations[5,6]. Here we used three of the largest neuroimaging datasets currently available—with a total sample size of around 50,000 individuals—to quantify BWAS effect sizes and reproducibility as a function of sample size. BWAS associations were smaller than previously thought, resulting in statistically underpowered studies, inflated effect sizes and replication failures at typical sample sizes. As sample sizes grew into the thousands, replication rates began to improve and effect size inflation decreased. More robust BWAS effects were detected for functional MRI (versus structural), cognitive tests (versus mental health questionnaires) and multivariate methods (versus univariate). Smaller than expected brain–phenotype associations and variability across population subsamples can explain widespread BWAS replication failures. In contrast to non-BWAS approaches with larger effects (for example, lesions, interventions and within-person), BWAS reproducibility requires samples with thousands of individuals.

MRI data (such as cortical thickness or resting-state functional connectivity (RSFC)) are increasingly being used for the ambitious task of relating individual differences in brain structure and function to typical variation in complex psychological phenotypes (for example, cognitive ability and psychopathology). To clearly distinguish such BWAS from other neuroimaging research, we formally define them as 'studies of the associations between common inter-individual variability in human brain structure/function and cognition or psychiatric symptomatology'. Classically univariate, BWAS have recently been facilitated by more powerful, but more difficult to interpret multivariate prediction techniques (for example, support vector regression (SVR) and canonical correlation analysis (CCA)). BWAS hold great promise for predicting and reducing

psychiatric disease burden and advancing our understanding of the cognitive abilities that underlie humanity's intellectual feats. However, obtaining MRI data remains expensive (approximately US$1,000 per hour), resulting in small-sample BWAS findings that have not been replicated[7–10].

Factors that have contributed to poor reproducibility of population-based research in psychology[11], genomics[12] and medicine[13], such as methodological variability[14], data mining for significant results[15], overfitting[16], confirmation and publication biases[17], and inadequate statistical power[5] probably also affect BWAS. Researchers are starting to address replication failures by standardizing analyses, pre-registering hypotheses, publishing null results and sharing data and code[18]. Nevertheless, there have been concerns that reliance on

relatively small samples (the median sample size (*n*) in openneuro.org studies as of September 2021 is 23) may also be contributing to BWAS replication failures[5,19–21]. Small studies are most vulnerable to sampling variability, the random variation of an association across population subsamples. Sampling variability decreases and associations stabilize with increasing sample sizes[19,22], at a rate of √*n*. Thus, if true brain-wide associations were smaller than previously assumed (for example, bivariate linear correlation *r* = 0.2–0.8), larger samples would be required to accurately measure them[19,20]. Other population-based sciences aiming to robustly characterize relatively small effects—such as epidemiology and genomics (that is, genome-wide association studies (GWAS))—have steadily increased sample sizes[12] from below 100 to over 1,000,000.

Recently, neuroimaging consortia have collected samples orders of magnitude larger than before (for example, the Adolescent Brain Cognitive Development[23] (ABCD) study, *n* = 11,874; Human Connectome Project[24] (HCP), *n* = 1,200; and UK Biobank[25] (UKB), *n* = 35,735), enabling accurate estimation of BWAS effect sizes. Beginning with the ABCD Study and using the HCP and UKB data for verification, we performed billions of univariate and multivariate analyses to evaluate BWAS effect sizes and reproducibility as a function of sample size, using sample sizes from small (*n* = 25) to large (*n* = 32,572).

## Precise BWAS require large samples

BWAS relate population variability in brain features (for example, RSFC between two brain regions (edge)) and behavioural phenotypes (for example, cognitive ability). To estimate brain-wide associations in ABCD data, we correlated widely used cortical thickness and RSFC metrics with 41 measures indexing demographics, cognition and mental health (Supplementary Table 1). Brain-wide associations were estimated across multiple levels of anatomical resolution in both structural (cortical vertices, regions of interest (ROI) and networks) and functional (connections (edges), principal components and networks) MRI data (Fig. 1). To ameliorate the effects of nuisance variables such as head motion, we applied strict denoising strategies (*n* = 3,928; >8 min; RSFC data post frame censoring at a filtered framewise displacement (filtered-FD) < 0.08 mm; Methods, 'DCANBOLDproc preprocessing'). Repeat analyses using less rigorous motion censoring that retained a larger subset of the full ABCD sample (*n* = 9,753), produced a similar BWAS effect size distribution (Supplementary Fig. 1).

BWAS analyses frequently link a single brain feature to a single behavioural phenotype. In Fig. 1a, b, we show the distributions of such univariate associations between cortical thickness and RSFC and two extensively studied phenotypes, cognitive ability (NIH Toolbox total score) and psychopathology (child behaviour checklist (CBCL) total score; Methods, 'Psychological and demographic data'; Supplementary Table 1; Supplementary Fig. 2 for non-overlapping histograms). In the full, rigorously denoised ABCD sample (*n* = 3,928), across all brain-wide associations, the median univariate effect size (|*r*|) was 0.01 (Extended Data Fig. 1). The top 1% largest of all possible brain-wide associations (around 11 million total associations) reached a |*r*| value greater than 0.06 (Fig. 1a, b). The top 10% largest associations were distributed across sensorimotor and association cortex (Fig. 1c, d). Across all univariate brain-wide associations, the largest correlation that replicated out-of-sample was |*r*| = 0.16. Sociodemographic covariate adjustment resulted in decreased effect sizes, especially for the strongest associations (top 1% Δ*r* = −0.014; Extended Data Fig. 2).

Smaller brain-wide association studies have reported larger univariate correlations (*r* > 0.2) than the largest effects we measured in much larger samples. To resolve this apparent contradiction, we simulated the effects of independent research groups using samples of varying sizes to estimate the same brain–phenotype association. For the strongest univariate brain-wide associations, we charted sampling variability as a function of sample size (Fig. 1e, f, *n* = 25–3,928). At *n* = 25, the 99% confidence interval for univariate associations was *r* ± 0.52,

documenting that BWAS effects can be strongly inflated by chance. In larger samples (*n* = 1,964 in each split half), the top 1% largest BWAS effects were still inflated by *r* = 0.07 (78%), on average (Supplementary Fig. 3). At *n* = 25, two independent population subsamples can reach the opposite conclusion about the same brain–behaviour association (for example, Fig. 1g, h), solely owing to sampling variability. See Supplementary Figs. 4–6 for sampling variability by sample size plots for all brain metrics and behavioural phenotypes.

Task fMRI data have also been correlated with cognitive phenotypes. Recent studies have suggested that treating task fMRI data similar to RSFC and combining the two modalities could strengthen BWAS effects slightly[26]. Therefore, we also estimated univariate BWAS associations for combined task and rest functional connectivity in ABCD Study data[27], which produced the same distribution of association strengths (top 1% |*r*| > 0.06) as RSFC. The HCP collected a wide variety of fMRI tasks, enabling us to compute all brain-wide associations between 86 task activation contrasts and 39 behavioural measures. The distributions of BWAS effect sizes for classical task fMRI activations and RSFC were closely matched (Extended Data Fig. 3, Supplementary Discussion).

Low measurement reliability can attenuate the observed correlation between two variables. Within-person measurement reliability for the exemplar behavioural phenotypes (NIH Toolbox[28], *r* = 0.90; CBCL[29], *r* = 0.94) and imaging measures (cortical thickness[30], *r* > 0.96; RSFC: ABCD, *r* = 0.48; HCP, *r* = 0.79; UKB, *r* = 0.39; Extended Data Fig. 4) are moderate to high. Whereas behavioural (NIH Toolbox, CBCL) and cortical thickness measures are already close to their reliability ceiling, further improvements in RSFC measurement reliability could theoretically increase effect sizes slightly (Supplementary Fig. 7, Supplementary Discussion). Theoretical maximum BWAS effect sizes are unlikely to be reached owing to fundamental biological limits on the strength of the true association and/or the limitations of behavioural phenotyping and MRI physics.

## Effect sizes replicate across datasets

Since the ABCD Study data (*n* = 11,874; age range: 9–10 years; 20 min, RSFC collected) were from a 21-site paediatric cohort (multiple scanner types), we sought to replicate BWAS effect sizes in single-site, single-scanner-type adult data. Thus, we used the HCP dataset which contains the most data per participant among large studies (*n* = 1,200; age range: 22–35 years; single scanner; 60 min, RSFC collected), and the UKB dataset which has the largest sample size, but less RSFC data per participant (*n* = 35,735; age range: 40–69 years; single scanner type; 6 min, RSFC collected), to verify univariate BWAS effect size distributions. All three datasets overlapped in containing RSFC and cognitive ability data. To control for sample size effects, the ABCD and UKB datasets were subsampled to match the HCP (*n* = 900, strict denoising). Across the three size-matched datasets we found similar effect size distributions for associations between RSFC and cognitive ability (Fig. 2; top 1% at *n* = 900 ABCD, |*r*| > 0.11; HCP, |*r*| > 0.12; UKB, |*r*| > 0.09; Extended Data Fig. 5; see Supplementary Fig. 8 for all ABCD/HCP cognitive measures).

To account for potential multi-site effects, we directly compared sampling variability between the HCP (single site) and ABCD datasets (Extended Data Fig. 6a), and between a single ABCD site (*n* = 603) and the 20 remaining sites (Extended Data Fig. 6b). Sampling variability was equivalent for single- and multi-site samples, underscoring the effectiveness of the ABCD Study's cross-site harmonization efforts[23]. The generalizability of the univariate BWAS effect size distribution (Fig. 2, Extended Data Figs. 5, 6) across age (9–69 years), sites, scanner types and pulse sequences suggests that it is universal to BWAS with current technologies and methods.

## Statistical errors limit reproducibility

Statistical error rates depend on effect sizes and significance testing thresholds. To quantify how the pairing of smaller than expected

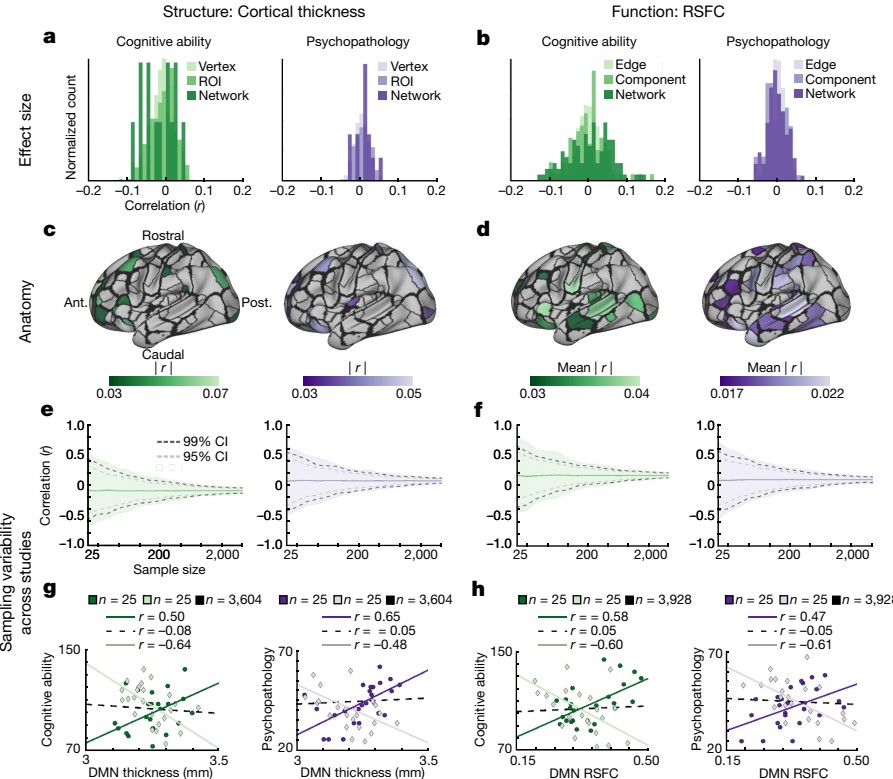

**Fig. 1 | Effect sizes and sampling variability of univariate brain-wide associations.** ABCD Study sample data (*n* = 3,928). **a**, **b**, Effect sizes were estimated using standard correlations (bivariate linear *r*). Brain-wide association histograms (normalized to per panel maximum bin) of cortical thickness with cognitive ability (left, green) and psychopathology (right, purple) at all levels of analysis (vertex, ROI and network; for separated levels of analysis see Supplementary Fig. 2a, b) (**a**), and RSFC with cognitive ability (left, green) and psychopathology (right, purple) at all levels of analysis (edge, network and component) (**b**). **c**, **d**, The largest brain-wide associations (ROI, top 10%) for cortical thickness with cognitive ability (left, green) and psychopathology (right, purple) (**c**), and RSFC with cognitive ability (left, green) and psychopathology (right, purple) (**d**). **e**, **f**, Sampling variability (1,000 resamples per sample size in logarithmically spaced bins: *n* = 25, 33, 50, 70, 100,

135, 200, 265, 375, 525, 725, 1,000, 1,430, 2,000, 2,800 and 3,604 (3,928 for cortical thickness)) of the largest brain-wide association for each brain–behavioural phenotype pair, for cortical thickness with cognitive ability (left, green) and psychopathology (right, purple) (**e**), and RSFC with cognitive ability (left, green) and psychopathology (right, purple) (**f**). Solid lines represent the mean across 1,000 resamples. Shading represents the minimum to maximum correlation range across subsamples, for a given sample size. Grey dashed line represents the 95% confidence interval and the black dashed line represents the 99% confidence interval. **f**, **g**, Examples of two *n* = 25 subsamples, in which inaccurate default mode network (DMN) correlations were observed for cortical thickness with cognitive ability (left, green) and psychopathology (right, purple) (**g**), and RSFC with cognitive ability (left, green) and psychopathology (right, purple) (**h**). Black dashed line denotes linear fit from full sample.

effect sizes and sampling variability (that is, random variation of an association across population subsamples) affects BWAS reproducibility, we used non-parametric bootstrapping[19] to generate smaller BWAS subsamples and characterized the relationship between statistical errors and sample size across significance thresholds ($P < 0.05$ to $P < 10^{-7}$; Fig. 3, Supplementary Fig. 9 for UKB) and verified the results with analytic statistical power estimations[31] (Supplementary Fig. 10).

Statistical errors were pervasive across BWAS sample sizes. Even for samples as large as 1,000, false negative rates (Fig. 3a) were very high (75–100%) and half of the statistically significant associations were inflated by at least 100% (Fig. 3b). More lenient statistical thresholding reduces false negatives and effect size inflation, but increases the rate of sign errors (Fig. 3c). Statistical power (1 − false negative rate), which indexes the probability of detecting a significant effect, remained low even for relatively large sample sizes: maximum statistical power 0.68 for *n* = 3,928 (Fig. 3d).

Given the high statistical error rates and low power of univariate BWAS in typically sized samples, we quantified the probability that a significant univariate association would replicate in a size-matched replication dataset (Fig. 3e; *P* from $10^{-7}$ to 0.05). In keeping with common practice, we defined successful replication as passing the same statistical threshold in sample and out of sample. At the largest split

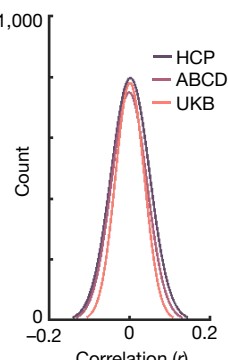

**Fig. 2 | Effect sizes of brain-wide associations are consistent across the largest neuroimaging study samples.** Univariate BWAS effect sizes from correlations (linear bivariate *r*) between fluid intelligence and edge-wise RSFC are shown for HCP, ABCD and UKB study samples. The ABCD (*n* = 3,928) and UKB (*n* = 32,572) datasets were subsampled (with replacement) 100 times to match the HCP sample size (*n* = 900), revealing consistent effect sizes (medians: HCP |*r*| = 0.03, ABCD |*r*| = 0.03, UKB |*r*| = 0.02). See Extended Data Fig. 5 for UKB resampling to both ABCD and HCP sample sizes.

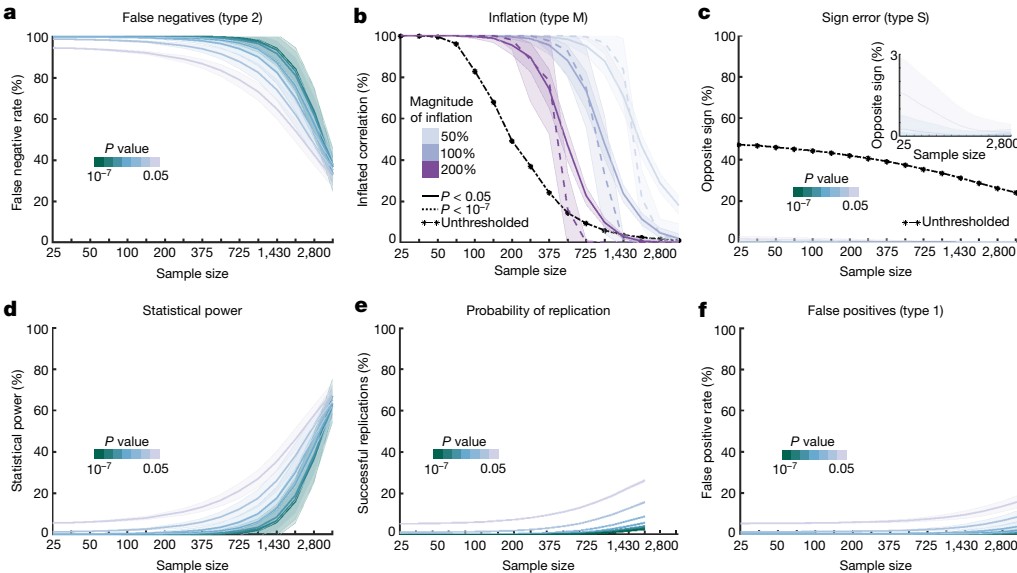

**Fig. 3 | Statistical errors and reproducibility of univariate brain-wide associations.** Data from ABCD Study sample ($n$ = 3,928; see Supplementary Fig. 9 for UKB). **a**, False negative rates (relative to full sample; see Methods, 'False positives, false negatives and power') for correlations (bivariate linear $r$) between psychological phenotypes and brain features (cortical thickness: vertex-wise; RSFC: edge-wise) as a function of sample size and two-tailed $P$ value ($P$ value thresholding was identical in full sample and subsamples; same $P$ values used in **c–f**). **b**, Magnitude error rates for three levels of effect size inflation (50%, 100% and 200%) as a function of sample size and statistical threshold ($P < 0.05$ and $P < 10^{-7}$ (Bonferroni-corrected); $P$ value threshold was

the same in full sample and subsamples). **c**, Sign error rates reported as the percentage of subsamples with the opposite sign as the full sample, as a function of sample size and $P$ value. **d**, Statistical power of subsamples relative to full sample (same sign, both significant) as a function of sample size and $P$ value. **e**, Probability of replicating (same sign, both significant) a univariate brain–phenotype association out-of-sample across $P$ values (note: data end at $n \approx 2,000$, as the replication sample is half of the full sample). Replication rates follow the square of power. **f**, False positive rates of subsamples relative to the full sample as a function of sample size and $P$ value.

half sample size ($n$ = 1,964), 25% of univariate BWAS replications succeeded with a threshold of $P < 0.05$. At sample sizes more typical for BWAS ($n < 500$), replication rates were around 5% (Fig. 3e).

Paradoxically, correcting for multiple comparisons reduces the probability of successfully replicating univariate BWAS effects (Fig. 3d, e). More stringent statistical thresholding reduces false positive rates (Fig. 3f) but increases false negative rates (Fig. 3a), thus lowering statistical power (Fig. 3d, Extended Data Fig. 7). In underpowered BWAS, stricter statistical thresholds select for very large correlations, which are the most likely to be inflated due to sampling variability (Fig. 1e, f). With Bonferroni multiple-comparisons correction ($P < 10^{-7}$), a sample size of 9,500 was required to be 80% powered for detecting the top 1% largest ($r > 0.06$) BWAS effects (Supplementary Fig. 10a), compared with a sample size of 2,200 for uncorrected $P < 0.05$ (Supplementary Fig. 10b).

## Multivariate BWAS reproducibility

Multivariate methods use weighted brain patterns to predict a single behavioural phenotype (SVR; for example, cognitive ability), or combinations of multiple phenotypes (CCA; for example, all NIH Toolbox subscales). To examine multivariate brain-wide associations as a function of sample size, we trained SVR (Supplementary Figs. 11–13) and CCA (Supplementary Figs. 14, 15) models on discovery set data (in-sample; including nested cross-validation (SVR) and principal component analysis (PCA) dimensionality reduction (SVR and CCA); Methods, 'Multivariate out-of-sample replication') and subsequently tested their generalization to the replication set using standard out-of-sample estimates of SVR ($r_{pred}$) and CCA ($r_{CV1}$) association strength (Fig. 4). Sampling variability was assessed by generating bootstrapped subsamples ($n$ = 100) for each sample size. Multivariate out-of-sample associations were tested for statistical significance using nonparametric null distributions (>99% confidence interval).

Across multivariate methods (SVR and CCA), imaging modalities (cortical thickness and RSFC), and behavioural phenotypes (cognitive ability and psychopathology), small discovery samples typical for neuroimaging generated variable, inflated in-sample associations that frequently did not pass statistical significance thresholds (Fig. 4a–d). Increasing sample sizes to thousands of participants provided moderate statistical replication with reduced variability and smaller differences between in-sample and out-of-sample associations. On average, RSFC (versus cortical thickness) and cognitive (versus psychopathology) measures provided stronger out-of-sample associations (Fig. 4a–d) that were closer to in-sample estimates (Fig. 4e). Narrowing the definition of replication to detecting statistical significance in out-of-sample data did not alleviate the need for large sample sizes (Supplementary Table 2).

Multivariate out-of-sample associations were stronger compared to univariate, particularly at large sample sizes (for example, maximum RSFC–crystallized intelligence association: SVR $r_{pred}$ = 0.39, univariate $r$ = 0.16). Even at the largest sample sizes ($n \approx 2,000$), multivariate in-sample associations remained inflated on average (in-sample to out-of-sample: $\Delta r$ = −0.29; Fig. 4e, Supplementary Fig. 16; see Extended Data Fig. 8 for univariate) and feature weights were variable (Supplementary Fig. 13). Out-of-sample replication was maximized by using a relatively low-dimensional feature space (Supplementary Figs. 11, 12, 14, 15), reaffirming that brain-wide associations are represented in widely distributed circuitry, consistent with univariate BWAS (Fig. 1c, d). Across behavioural phenotypes, multivariate out-of-sample associations were robustly linked to univariate effect sizes ($r$ = 0.79, $P < 0.001$; Fig. 4f).

## The underpowered BWAS paradox

At smaller sample sizes, the largest, most inflated BWAS effects are most likely to be statistically significant and therefore, paradoxically,

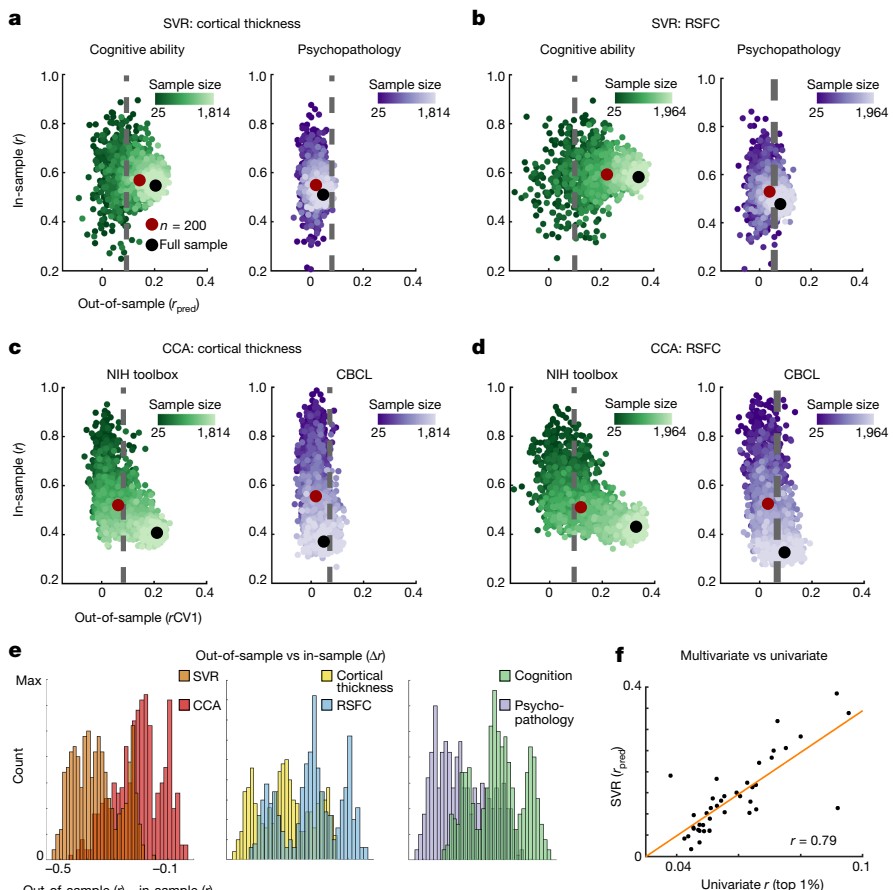

**Fig. 4 | Multivariate brain-wide associations. a–f**, In-sample brain–behavioral phenotype associations as a function of out-of-sample associations and sample size. Mean multivariate brain–behavioral phenotype associations across 100 bootstrap samples at $n = 200$ (red dots) and for the full sample (black dots). Grey dashed lines represent the significance threshold for out-of-sample correlations (>99% confidence interval of permutations), determined on the full sample (see Methods, 'Multivariate out-of-sample replication'). Data are from the ABCD Study; full sample sizes: cortical thickness $n = 1,814$; RSFC $n = 1,964$. **a**, **b**, For SVR, out-of-sample association strength is reported as the correlation between predicted and observed phenotype scores ($r_{pred}$) using models trained on the discovery set. SVR of cortical thickness (**a**) and RSFC (**b**), with cognitive ability (green, left) and psychopathology (purple, right). **c**, **d**, For CCA, out-of-sample association strength is reported as the correlation of

phenotypic and brain scores in the first canonical variate pair ($r_{CV1}$) when discovery set weights are applied to the replication set. CCA of cortical thickness (**c**) and RSFC (**d**), with all NIH Toolbox (green, left) and CBCL (purple, right) subscales. **e**, Differences between out-of-sample (SVR: $r_{pred}$; CCA: $r_{CV1}$) and corresponding in-sample associations by multivariate method (left), imaging modality (middle) and behavioural phenotype (right); normalized to per-panel maximum. On average, out-of-sample associations (mean $r = 0.17$) were smaller ($\Delta r = -0.29$; 63% reduction) than in-sample associations (mean $r = 0.46$), similar to replication effect size reductions in cancer biology[49] and psychology[50]. **f**, SVR out-of-sample association ($r_{pred}$) as a function of univariate effect size ($r$; top 1% for each phenotype) across the 41 phenotypes (bivariate linear $r = 0.79$, orange line).

the most likely to be published[5,21,32]. Typically, BWAS have been sufficiently powered to only detect statistical significance for inflated associations (Fig 3d). High sampling variability in smaller samples frequently generates strong associations by chance[19] (Fig. 1e, f). Stricter in-sample statistical thresholding (that is, multiple-comparison correction)—which is common in neuroimaging—lowers BWAS power, thus trapping us deeper in the paradox by selecting for even more inflated effects (Fig. 3). When attempting to replicate inflated BWAS associations, regression to the mean (actual effect size) makes non-significance (that is, replication failure) the most likely outcome (Figs. 3, 4, Extended Data Fig. 8). Bias in favour of significant, larger BWAS effects has limited the publication of null results, perpetuating inflated effect sizes that form the basis for subsequent power and meta analyses.

## Importance of small-sample neuroimaging

There is no one-size-fits-all solution for neuroimaging studies; minimum sample size requirements depend on the study design.

Neuroimaging-only studies are typically adequately powered at small sample sizes. For example, central tendencies of human functional brain organization among groups can be accurately represented by averaging within small samples (that is, $n = 25$; Supplementary Fig. 17). Precise individual-specific RSFC and fMRI activation brain maps can be generated by repeatedly sampling the same individual[33]. Small samples have also provided blueprints for reducing MRI artefacts[34], increasing the amount of usable data[35].

Using non-BWAS approaches, many fundamental links between the human brain and behaviour have been uncovered and replicated in small neuroimaging samples[36]. Within-person designs (for example, longitudinal[37]), studies with induced effects (for example, lesions[38] or tasks[39]), or both (for example, interventions[40]) frequently have increased measurement reliability and effect sizes. For rarer clinical conditions, amassing large samples is impossible. In many cases, within-person, induced-effects approaches are not only cost-effective, but also most relevant to clinical care. Thus, small-sample neuroimaging will always be critical for studying the human brain.

## Importance of large samples for BWAS

Large neuroimaging consortium data (ABCD, HCP and UKB) have revealed that small BWAS effects and population sampling variability routinely results in inflated, irreproducible brain–phenotype associations until sample sizes reach well into the thousands (Extended Data Fig. 9). Therefore, BWAS should use datasets with at least thousands of high-quality, standardly processed samples[14]. Additional consideration should be given to potential confounding effects and interpretations of statistical significance[41].

The recovery of genomics from its reproducibility crisis has set a valuable example for BWAS[12]. Early candidate-gene studies were underpowered and many associations between common genetic variants and psychiatric phenotypes could not be replicated[42]. In response, GWAS consortia have grown genomic samples into the millions[43] and taken advantage of specialized study designs (for example, twins) and methodological innovations (for example, polygenic risk scores) and set strict data standards. Fortunately, BWAS findings can achieve reproducibility in relatively smaller samples than GWAS, owing to larger effect sizes.

## Reproducibly linking brain and behaviour

All brain–behaviour studies will benefit from technological advances that generate higher quality brain and behavioural data with greater efficiency, such as real-time quality control[35], multi-band multi-echo[44] sequences and thermal denoising for fMRI[45], as well as deep behavioural phenotyping with ecological momentary assessment[46] and passive sensing.

As with GWAS[47], funding agencies should boost the aggregation of BWAS-appropriate datasets through mandatory sharing policies. Even for large datasets collected and processed identically, in-sample associations are stronger than out-of-sample replications (Fig. 4e, Extended Data Fig. 8); therefore, reporting both in-sample and out-of-sample effect sizes should be a requirement for publication and funding. BWAS may also benefit from focusing data collection on the most robust brain–phenotype associations (for example, functional versus structural and direct behavioural versus questionnaire).

The brain, in contrast to the genome, is expected to change over time and can be manipulated ethically. For greater effect sizes and statistical power, neuroscience should focus on within-participant study designs over cross-sectional study designs, and on interventional (therapy, medications, brain stimulation and surgery) over observational study designs. Rather than associating pre-defined psychological constructs and brain features[48], data-driven, combined brain–behaviour phenotypes will further advance our understanding of cognition and mental health. Altogether, our prospects for linking neuroimaging markers to complex human behaviours are better than ever.

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

[1]Department of Psychiatry, Washington University School of Medicine, St Louis, MO, USA. [2]Department of Psychiatry, Massachusetts General Hospital, Harvard Medical School, Boston, MA, USA. [3]Department of Psychology, University of Pittsburgh, Pittsburgh, PA, USA. [4]Department of Psychiatry, University of Pittsburgh, Pittsburgh, PA, USA. [5]Department of Bioengineering, University of Pittsburgh, Pittsburgh, PA, USA. [6]Department of Neurology, Washington University School of Medicine, St Louis, MO, USA. [7]University of Minnesota Informatics Institute, University of Minnesota, Minneapolis, MN, USA. [8]Department of Psychology, University of Minnesota, Minneapolis, MN, USA. [9]Masonic Institute for the Developing Brain, University of Minnesota Medical School, Minneapolis, MN, USA. [10]Department of Pediatrics, University of Minnesota Medical School, Minneapolis, MN, USA. [11]Department of Psychiatry, Oregon Health and Science University, Portland, OR, USA. [12]Minnesota Supercomputing Institute, University of Minnesota, Minneapolis, MN, USA. [13]Department of Electrical and Computer Engineering, National University of Singapore, Singapore, Singapore. [14]Centre for Sleep and Cognition, National University of Singapore, Singapore, Singapore. [15]Centre for Translational MR Research, National University of Singapore, Singapore, Singapore. [16]N.1 Institute for Health, Institute for Digital Medicine, National University of Singapore, Singapore, Singapore. [17]Department of Biomedical Engineering, Washington University in St Louis, St Louis, MO, USA. [18]Department of Cognitive Science, University of California San Diego, La Jolla, CA, USA. [19]Department of Radiology, Washington University School of Medicine, St Louis, MO, USA. [20]Department of Neurological Surgery, Washington University School of Medicine, St Louis, MO, USA. [21]Department of Psychological and Brain Sciences, Washington University in St Louis, St Louis, MO, USA. [22]Department of Psychiatry, University of Vermont, Burlington, VT, USA. [23]Division of Biostatistics, University of California San Diego, La Jolla, CA, USA. [24]Oxford Big Data Institute, Li Ka Shing Centre for Health Information and Discovery, Nuffield Department of Population Health, University of Oxford, Oxford, UK. [25]Integrative Sciences and Engineering Programme, National University of Singapore, Singapore, Singapore. [26]Martinos Center for Biomedical Imaging, Massachusetts General Hospital, Charlestown, MA, USA. [27]Institute of Child Development, University of Minnesota Medical School, Minneapolis, MN, USA. [28]Program in Occupational Therapy, Washington University School of Medicine, St Louis, MO, USA. [29]Department of Pediatrics, Washington University School of Medicine, St Louis, MO, USA. [30]These authors contributed equally: Scott Marek, Brenden Tervo-Clemmens. [31]These authors jointly supervised this work: Damien A. Fair, Nico U. F. Dosenbach. [✉]e-mail: smarek@wustl.edu; btervo-clemmens@mgh.harvard.edu; faird@umn.edu; ndosenbach@wustl.edu

# Methods

## ABCD Study sample

This project used the baseline ABCD BIDS (Brain Imaging Data Structure) data consisting of RSFC data from 10,259 participants released through the ABCD-BIDS Community Collection[51] (ABCD collection 3165; https://github.com/ABCD-STUDY/nda-abcd-collection-3165) and demographic and behavioural data from 11,572 9–10 year old participants from the ABCD 2.0 release[52]. The ABCD Study obtained centralized institutional review board (IRB) approval from the University of California, San Diego. Each of the 21 sites also obtained local IRB approval. Ethical regulations were followed during data collection and analysis. Parents or caregivers provided written informed consent, and children gave written assent.

In addition to data from the ABCD 2.0 release, we used the ABCD reproducible matched samples[51] (ARMS), available in ABCD collection 3165, that divided individuals from the full behavioural sample ($n = 11,572$) into discovery ($n = 5,786$) and replication ($n = 5,786$) sets, which were matched across 9 variables: site location, age, sex, ethnicity, grade, highest level of parental education, handedness, combined family income, and prior exposure to anaesthesia. Family members (that is, sibling pairs, twins and triplets) were kept together in the same set and the two sets were matched to include equal numbers of single participants and family members. These split ARMS datasets were used for replicability analyses.

Head motion can systematically bias neuroimaging studies[53]. However, these systematic biases can be addressed through rigorous head motion correction. Therefore, we used strict inclusion criteria with regard to head motion. Specifically, inclusion criteria for the current project (see Casey et al.[23] for broader ABCD inclusion criteria) consisted of at least 600 frames (8 min) of low-motion[54] (filtered-FD < 0.08) RSFC data. Our final dataset consisted of RSFC data from a total of $n = 3,928$ youth across the discovery ($n = 1,964$) and replication ($n = 1,964$) sets. The final discovery and replication sets did not differ in mean framewise displacement (difference in means = 0.002, $t = 0.60$, $P = 0.55$) or total frames included (difference in means = 6.4, $t = 0.94$, $P = 0.35$). The participant lists for ARMS samples can be found in the ABCD-BIDS Community Collection (ABCD collection 3165) for community use[51].

## ABCD MRI acquisition

Imaging was performed at 21 sites in the United States, harmonized across Siemens Prisma, Philips and GE 3T scanners. Details on image acquisition can be found in ref. [23]. Twenty minutes (4 × 5 min runs) of eyes-open resting-state blood oxygenation level dependent (BOLD) data were acquired to ensure at least 8 min of low-motion data. All resting-state scans fMRI scans used a gradient-echo echo planar imaging (EPI) sequence (repetition time = 800 ms, echo time = 30 ms, flip angle = 90°, voxel size = 2.4 mm³, 60 slices). Head motion was monitored using framewise integrated real-time MRI monitoring (FIRMM) software at many of the Siemens sites[35].

## ABCD-BIDS processing overview

ABCD and UKB MRI data processing was completed with the freely available ABCD-BIDS pipeline[51] (https://github.com/DCAN-Labs/abcd-hcp-pipeline). Data were downloaded and converted to the BIDS format using ABCD-Dicom2BIDS (https://github.com/DCAN-Labs/abcd-dicom2bids). Only data that passed the fast-track quality control (QC; tagged prior to ABCD release 2.0) were processed (also see release notes: https://collection3165.readthedocs.io/en/stable/). The ABCD-BIDS pipeline is a modification of the original HCP pipeline[55]. In brief, this MRI data-processing pipeline comprises six stages. (1) PreFreesurfer normalizes anatomical data. This normalization entails brain extraction, denoising, and then bias field correction on anatomical T1 and/or T2 weighted data. The ABCD-HCP pipeline includes two additional modifications to improve output image quality.

ANTs[56] DenoiseImage models scanner noise as a Rician distribution and attempts to remove such noise from the T1 and T2 anatomical images. Additionally, ANTs N4BiasFieldCorrection attempts to smooth relative image histograms in different parts of the brain and improves bias field correction. (2) FreeSurfer[57] constructs cortical surfaces from the normalized anatomical data. This stage performs anatomical segmentation, white–grey and grey–CSF cortical surface construction, and surface registration to a standard surface template. Surfaces are refined using the T2 weighted anatomical data. Mid-thickness surfaces, which represent the average of white–grey and grey–CSF surfaces, are generated here. (3) PostFreesurfer converts prior outputs into an HCP-compatible format (that is, CIFTIs) and transforms the volumes to a standard volume template space using ANTs nonlinear registration, and the surfaces to the standard surface space via spherical registration. (4) The Vol (volume) stage corrects for functional distortions via reverse-phase encoding spin-echo images. All resting-state runs underwent intensity normalization to a whole-brain-mode value of 1,000, within run correction for head movement, and functional data registration to the standard template. Atlas transformation was computed by registering the mean intensity image from each BOLD session to the high resolution T1 image, and then applying the anatomical registration to the BOLD image. This atlas transformation, mean field distortion correction, and resampling to 3 mm³ atlas space were combined into a single interpolation using the FSL[58] applywarp tool. (5) The Surf (surface) stage projects the normalized functional data onto the template surfaces, as described below. (6) We have added an fMRI and fcMRI preprocessing stage, DCANBOLDproc, also described below. (7) Last, an executive summary is provided for easy participant-level QC across all processed data.

## fMRI surface processing

The BOLD fMRI volumetric data were sampled to each participant's original mid-thickness left and right-hemisphere surfaces constrained by the grey-matter ribbon. Once sampled to the surface, time courses were deformed and resampled from the individual's original surface to the 32 k fs_LR surface in a single step. This resampling allows point-to-point comparison between each individual registered to this surface space. These surfaces were then combined with volumetric subcortical and cerebellar data into the CIFTI format using Connectome Workbench[59], creating full brain time courses excluding non-grey matter tissue. Finally, the resting-state time courses were smoothed with a 2 mm full-width-half-maximum kernel applied to geodesic distances on surface data and euclidean distances on volumetric data.

## DCANBOLDproc preprocessing

Additional BOLD preprocessing steps were executed to reduce spurious variance unlikely to reflect neuronal activity[34]. First, a respiratory filter was used to improve framewise displacement estimates calculated in the Vol stage[54]. Second, temporal masks were created to flag motion-contaminated frames using the improved framewise displacement estimates[53]. Frames with a filtered-FD > 0.3 mm were flagged as motion-contaminated for nuisance regression only. After computing the temporal masks for high motion frame censoring, the data were processed with the following steps: (1) demeaning and detrending, (2) interpolation across censored frames using least squares spectral estimation of the values at censored frames so that continuous data can be (3) denoised via a GLM with whole brain, ventricular, and white matter signal regressors, as well as their derivatives. Denoised data were then passed through (4) a band-pass filter (0.008 Hz < $f$ < 0.1 Hz) without re-introducing nuisance signals[60] or contaminating frames near high-motion frames.

## Generation of RSFC matrices

ABCD RSFC data consists of 4 × 5 min runs. For each participant with full brain coverage, all available RSFC data were concatenated and high

motion frames (filtered-FD > 0.08) were censored. The timeseries of BOLD activity for each ROI was correlated to that of every other ROI (333 cortical ROIs from Gordon et al.[61]; 61 subcortical ROIs from Seitzman et al.[62]), forming a 394 × 394 correlation matrix, which was subsequently Fisher $z$-transformed. For network level analyses, correlations were averaged across previously defined canonical functional networks[61]. Inter-individual difference connectome-wide spatial components, which are not bound by network boundaries[63,64], were computed by performing PCA on a matrix composed of all ROI × ROI pairs (edges) from each participant.

## Generation of cortical thickness metrics

For each participant, cortical thickness was extracted from 59,412 cortical vertices. For ROI level matrices, cortical thickness was averaged within each cortical parcel[61] ($n = 333$). For network level matrices, cortical thickness was averaged within each cortical network[61] ($n = 13$). Inter-individual spatial components were computed by performing PCA on a matrix composed of all cortical vertices from each participant.

## Psychological and demographic data

The ABCD Study population is well-characterized with hundreds of demographic, physical, cognitive, and mental health variables[65]. The current project examined the associations between 41 of these variables (Supplementary Table 1) and brain structure (cortical thickness) and function (RSFC). Psychological and demographic variables were selected to reflect the primary domains of interest, cognition (individual subscales and composite scores from the NIH Toolbox) and mental health (individual subscales and composite scores from the CBCL), as well as demographic and physical variables relevant to development (for example, age) and health (for example, body mass index).

## Psychological and demographic covariates

The primary goal of this project was to study how the pairing of brain–phenotype effect sizes and sampling variability (random variation across samples, as opposed to systematic variation threatening causal inference[66]) can account for wide-spread replication failures. Hence, our results focus on bivariate associations (correlation) and standard multivariate models linking brain structure and function to psychological and demographic variables without covariate adjustment. However, we did examine the influence of sociodemographic covariates standardly used in ABCD analyses (race, gender, parental marital status, parental income, Hispanic versus non-Hispanic ethnicity, family and data collection site) on BWAS effect sizes noting that they generally decrease effect sizes, particularly for the largest BWAS effects (see Extended Data Fig. 2). Furthermore, the ABCD subsamples (ARMS; see above) we used for replication analyses are matched for salient demographic factors (site location, family composition, age, sex, ethnicity, grade, highest level of parental education, handedness, combined family income and prior exposure to anaesthesia; see above). Also, where possible, ABCD-distributed age-corrected scores were used, given (1) well-established age-related changes in these measures and (2) age-corrected scores improved normality for many measures (for example, CBCL syndrome scales and broadband factors).

## Capture of psychological and demographic data

The ABCD Data Analysis and Informatics Center (DAIC) has released an online tool called DEAP (Data Exploration and Analysis Portal), which can be accessed at https://deap.nimhda.org/. In this Article, we introduce an additional tool called ABCDE (ABCD Boolean Capture Data Explorer, developed by B.P.K.), which we have used for preparation of the data herein. ABCDE complements DEAP by allowing for finer-grained control of data extraction on the researcher's own computer rather than through a web portal. The source code and documentation can be accessed at https://gitlab.com/Dosenbach-Greene/abcde.

## Univariate brain–behavioural phenotype correlations

For each brain measure at a given level of organization, we correlated the brain measures (structure: cortical thickness; function: RSFC) with each psychological variable. Cognitive ability (total composite score on the NIH Toolbox) and psychopathology (total score on the CBCL) are presented in the main text; all others are included in the Extended Data Fig. 1. Correlations between brain and phenotypes were generated for RSFC at the edge level (ROI–ROI pair ($n = 77,421$)), network level (average of RSFC within and between each network ($n = 105$)) and component level (principal component weights ($n = 100$)). To extract components representing inter-individual differences, we vectorized each participant's RSFC matrix, concatenated the vectorized matrices and then performed PCA (Matlab's pca.m function). Correlations between brain and phenotypes were generated for cortical thickness at the vertex level ($n = 59,412$), ROI level ($n = 333$) and network level ($n = 13$). Repeat analyses employing less rigorous motion censoring and thus retaining a larger subset of the full ABCD sample ($n = 9,753$) replicated the effect sizes (top 1% largest effects: $|r| > 0.06$).

## Resampling procedures

To examine the distribution of correlations for iteratively larger sample sizes, we randomly selected participants with replacement from the full sample ($n = 3,928$, post denoising) at logarithmically spaced sample sizes (16 intervals: $n = 25, 33, 50, 70, 100, 135, 200, 265, 375, 525, 725, 1,000, 1,430, 2,000, 2,800$ and $3,928$). For cortical thickness data, the full sample contained the same sampling bins, with the exception of the final bin (full sample), which contained $n = 3,604$ participants. At each sample size, we randomly sampled participants 1,000 times, resulting in 16,000 brain–psychological phenotype resamplings for each brain–phenotype correlation. For multivariate approaches, 100 bootstrap samples were computed across the logarithmically spaced sample sizes (16 intervals: $n = 25, 33, 45, 60, 80, 100, 145, 200, 256, 350, 460, 615, 825, 1,100, 1,475$ and $1,964$ ($1,814$ for cortical thickness)). We note that the iterations were reduced for multivariate methods (100 iterations) owing to their high computational costs. In addition, the multivariate analyses were primarily focused on mean estimates, rather than the full distribution. We also performed sensitivity analyses to quantify sampling variability using data from only singletons (that is, no sibling and/or twin pairs), which was nearly identical to sampling variability in the full sample (included siblings and/or twins; Extended Data Fig. 10; $\Delta r = 0.0005$). For highlighting the effects of sampling variability (Fig. 1e, f), we extracted the brain–phenotype correlation with the largest effect size for each imaging modality (cortical thickness and RSFC) and exemplar phenotype (cognitive ability and psychopathology). The sampling variability (range of possible correlations, 99% confidence interval and 95% confidence interval) at each sampling interval for correlations between RSFC and cortical thickness with cognitive ability and psychopathology are presented in the main text (Fig. 1e, f); correlations between brain measures and other behaviours can be found in Supplementary Figs. 4, 5.

## Sampling variability examples with a sample size of 25

Using the outputs from the resampling procedures above, we used the 1,000 resamplings with $n = 25$ to examine the correlation between the DMN and cognitive ability (total composite score on the NIH Toolbox), as well as the DMN and psychopathology (total problem score on the CBCL), for both cortical thickness and RSFC. To demonstrate how sampling variability affects correlations, the 1,000 resamples were ranked by effect size. Subsequently, we selected two samples from the top 10 samples (in terms of effect size); one with a significant positive association and one with a significant negative association.

## ABCD task data

Data from three in-scanner fMRI tasks (n-back, stop signal, monetary incentive delay) were concatenated to the 4 × 5 min resting-state runs (rest + task) to determine whether additional data affected the effect size estimates. After data were concatenated across the 4 conditions (rest + 3 task states), correlation matrices were generated and correlated with psychological phenotypes as detailed above, under univariate brain–behavioural phenotype correlations. Task events were not regressed[67]. Data processing steps for task data were the same as RSFC, including the removal of frames with a filtered-FD > 0.08 mm.

## Correlations between behavioural phenotypes

To examine the range of sampling variability as a function of sample size between 41 psychological and demographic measures (Supplementary Fig. 6), we randomly selected participants with replacement from the full behavioural sample ($n = 11,572$) at logarithmical spaced sample sizes (9 intervals: $n = 25, 50, 100, 200, 500, 1,000, 2,000, 4,000$ and $9,000$). At each interval, we randomly sampled participants 1,000 times, resulting in 9,000 behaviour–behaviour phenotype correlation resamplings for each association. For each association between behavioural phenotypes, we quantified sampling variability at each sampling bin as the range of correlations observed through this resampling procedure.

## False positives, false negatives and power

False negative (Fig. 3a) and false positive (Fig. 3f) rates were derived through resampling (see 'Resampling procedures') for all edge-wise brain-wide associations. For each sample size bin (16 total), we randomly sampled with replacement $n$ individuals (1,000 subsamples) and computed the brain–behavioral phenotype correlation and associated $P$ value. A correlation was deemed significant if it passed a threshold ($P$ value range: $<0.05$ to $<10^{-7}$ (Bonferroni-corrected) across 77,421 ROI–ROI pairs) in the full sample (cortical thickness $n = 3,604$, RSFC $n = 3,928$). At each sample size, if a correlation in the full sample was not significant, we determined the percentage of studies that resulted in a false positive significant correlation across a broad range of $P$ values ($0.05$ to $10^{-7}$). Conversely, if a correlation in the full sample was significant ($P < 0.05$ to $10^{-7}$), we determined the percentage of studies that resulted in a false negative non-significant correlation across a broad range of $P$ values ($10^{-7}$ to $0.05$). Statistical power (Fig. 3d) was calculated as $1 -$ false negative rate.

## BWAS correlation inflation

For each univariate brain-wide association in the full sample (cortical thickness $n = 3,604$; RSFC $n = 3,928$) at the vertex/edge level, we determined whether or not a correlation was significant (using two-tailed $P < 0.05$ (uncorrected) and $P < 10^{-7}$ (Bonferroni corrected for multiple comparisons) thresholds). Then, for each significant correlation in the full sample, we extracted all of the significant correlations ($P < 0.05$ and $10^{-7}$) observed across 1,000 subsamples at each sample size bin. Of these significant correlations in subsamples at each sample size bin, we determined the percentage that were inflated, relative to the full sample effect size, across varying magnitudes ($50\%, 100\%$ and $200\%$; Fig. 3b).

## BWAS sign errors

Each brain-wide association was extracted from the full sample as a reference. Across the 1,000 subsamples within a sampling bin, we determined the percentage of correlations that had the opposite correlation sign as the correlation sign in the full sample, thresholding the subsamples at the same $P$ values as all other analyses of statistical errors ($P < 10^{-7}$ to $0.05$).

## Univariate BWAS replication

Replication is commonly defined as detecting a significant association (for example, $P < 0.05$) that was deemed significant ($P < 0.05$) in a previous sample (Fig. 3e). To determine the probability of replicating a brain–phenotype association in a new data (out-of-sample) at a given sample size, we correlated every brain feature (RSFC edge, cortical thickness vertex) with each behavioural phenotype in 1,000 bootstrapped samples across sample sizes (same sampling bins as listed under 'Resampling procedures'). For each behavioural phenotype, sample size ($n = 25, 33, 45, 60, 80, 100, 145, 200, 256, 350, 460, 615, 825, 1,100, 1,475$ and $1,964$ ($1,814$ for cortical thickness); note: data end at $n \approx 2,000$ as the replication sample is half of the full), and bootstrapped subsample, we first determined the brain–behavioural phenotype associations that were significant (at $P < 10^{-7}$ to $0.05$) in the discovery (in-sample) dataset. Next, we extracted the same brain features from the replication (out-of-sample) dataset and quantified the percentage of associations that were also significant in the replication dataset. Note, to mirror a process of replicating existing effects, we used the number of identified significant associations in the discovery sample as the total number of features that could be replicated (as opposed to the total number of brain features regardless of discovery sample significance). For example, if all significant BWAS in the discovery sample were also significant in the replication sample, the probability of replication would be 100%.

## Effect sizes in HCP replication

We used data from $n = 900$ individuals from the HCP 1,200 Subject Data Release (aged 22–35 years). All HCP participants provided informed consent. A custom Siemens SKYRA 3.0T MRI scanner and a custom 32-channel head matrix coil were used to obtain high-resolution T1-weighted (MP-RAGE, TR = 2.4 s, 0.7 mm$^3$ voxels) and BOLD contrast sensitive (gradient-echo EPI, multiband factor 8, TR = 0.72 s, 2 mm$^3$ voxels) images from each participant. The HCP used sequences with left-to-right (LR) and right-to-left (RL) phase encoding, with a single RL and LR run on each day for two consecutive days for a total of four runs[68]. MRI data were preprocessed as previously described[62]. All HCP data are available at https://db.humanconnectome.org/.

Similar to the ABCD data, we extracted the timeseries from a total of 394 cortical and subcortical ROIs, correlated and Fisher $z$-transformed them. Data from the NIH Toolbox were correlated with each edge of the RSFC correlation matrix across participants. Across all NIH Toolbox subscales, the tails of the distributions of the resulting brain–behavioural phenotype correlations were compared to 100 subsampled ABCD brain–behavioural phenotype correlations ($n = 877$, matching HCP sample size). In Supplementary Fig. 8, we show the distributions of brain–behavioural phenotype correlations for ABCD and HCP data, for each NIH Toolbox subscale.

## Effect sizes in UKB replication

We used pre-processed resting-state data from $n = 32,572$ individuals from the January 2020 UKB release[69], processed with the same processing pipeline as the ABCD data. All UKB participants provided informed consent. For a complete description of study flow and imaging protocols, see Littlejohns et al.[70]. The UKB collects measures of fluid intelligence, which we used to correlate with RSFC, mimicking ABCD and HCP samples. For Fig. 2, we used $100 \times n = 900$ subsamples from the ABCD and UKB datasets to match the sample size of HCP for the associations between RSFC and fluid intelligence ($n = 900$). We subsequently determined the threshold to reach the top 1% strongest RSFC associations with fluid intelligence in each of the three datasets.

## Sampling variability in HCP replication

To quantify the degree of sampling variability in single site, single scanner HCP data compared to multi-site, multi-scanner ABCD data, we subsampled ABCD RSFC data to match HCP sample sizes ($n = 877$, denoised and complete behavioural data across all NIH Toolbox subscales). For each dataset, we carried out resampling, as detailed under 'Resampling procedures' (12 intervals: $n = 25, 33, 50, 70, 100, 135, 200,$

265, 375, 525, 725 and 875), across all NIH Toolbox subscales. The range of correlations and 95% confidence interval observable in each sampling bin are shown in Extended Data Fig. 6a for both HCP and ABCD data.

### Sampling variability for single-site ABCD versus multi-site ABCD
We directly compared single site ABCD data (site 16; $n = 603$) with multi-site ABCD data ($n = 3,325$, 20 sites−site 16 was excluded) using 1,000 bootstrapped samples at 10 sample size intervals: $n = 25, 33, 50, 70, 100, 135, 200, 265, 375$ and $525$. For this analysis, we used the associations between RSFC and all NIH Toolbox subscales (Extended Data Fig. 6b). The range of correlations and 95% confidence interval in each resampling bin is shown in Extended Data Fig. 6b for single-site and multi-site ABCD data.

### BWAS effect sizes in task activation versus RSFC in HCP
We estimated the effect sizes between task activations (86 total contrasts, see Supplementary Table 3) and behavioural phenotypes[71] (39 total, see Supplementary Table 4) across three levels of analysis: vertices, ROIs and networks ($n = 844$). In these same individuals, we estimated the effect sizes between RSFC and the same phenotypes across three levels of analysis: edges, principal components, and networks. To compare the resulting effect size distributions (for example, Extended Data Fig. 3), we determined the top 1% strongest effect sizes, as well as the maximum correlation (absolute value).

### Multivariate out-of-sample replication
For multivariate out-of-sample replication, we used SVR and CCA. SVR with a linear kernel was performed using the e1071 package in the R environment (version 3.5.2) to predict primary phenotypes (psychopathology and cognitive ability) and other demographics and psychological phenotypes (Supplementary Figs. 11, 12) from individual differences in either RSFC or cortical thickness. One hundred bootstrap samples (sampling with replacement) were generated for each sample size. Hyperparameter tuning was examined in (1) split halves of the full discovery sample for multiple cognitive (NIH Toolbox) and psychopathology (CBCL) scales and (2) tenfold cross-validation within the full discovery sample for primary phenotypes (psychopathology and cognitive ability; Supplementary Figs. 11, 12). Hyperparameter tuning did not appreciably change out-of-sample prediction estimates to the replication sample (for example, average out-of-sample correlation difference between tuned and non-tuned models: RSFC = −0.006, cortical thickness = 0.014; Supplementary Figs. 11, 12). Figure 4a, b use default hyperparameters and PCA dimensionality reduction (with a threshold of 50% variance explained in the discovery set, for each sample size) prior to SVR, given that this procedure balanced out-of-sample prediction and model complexity for nearly all model types (Supplementary Figs. 11, 12). Replication set data were not used to estimate principal components, but rather replication set data were projected into component space via independently estimated loading matrices for each subsample of the discovery set to prevent bias. An alternative strategy of univariate feature ranking was also examined, where SVR models were trained on the 5,000, 10,000 or 15,000 vertices (cortical thickness) or edges (RSFC) with the highest bivariate correlation to the variable of interest in the training dataset, but this approach resulted in lower out-of-sample prediction (Supplementary Figs. 11, 12). Out-of-sample association strength is reported as the correlation between predicted and observed phenotypic scores ($r_{pred}$; using models trained on the discovery set). Significance thresholds for out-of-sample replication ($r_{pred}$) were estimated via permutation testing (1,000 iterations) with models trained on the full discovery set (RSFC: $n = 1964$; cortical thickness: $n = 1,814$) and tested on the full replication set.

CCA was performed using Matlab's (2019A) cannoncor.m function for joint associations of the NIH Toolbox and CBCL with individual differences in either RSFC or cortical thickness. Equivalent bootstrapping and subsampling of the in-sample discovery set were tested and applied to the out-of-sample replication set, as in the SVR analyses. To model sampling variability across sample sizes, 100 bootstrap (sampling with replacement) samples were generated for each sample size. As with SVR, Fig. 4c, d used PCA dimensionality reduction (threshold of 20% variance explained in the in-sample discovery set, for each sample size) prior to CCA given that this maximized out-of-sample correlation ($r_{CV1}$; Supplementary Figs. 14, 15). CCA models were fit on iteratively larger subsamples of the in-sample discovery dataset. The first canonical vector (CV1) weights were extracted and applied to the full out-of-sample brain and behavior data. This resulted in the out-of-sample correlation ($r_{CV1}$) between multivariate brain and behavior data. Significance thresholds for out-of-sample replication were estimated via permutation testing (1,000 iterations) with models trained on the full ABCD discovery set (RSFC: $n = 1964$; cortical thickness: $n = 1,814$) and tested on the full replication set.

### Towards a new era of BWAS
In Extended Data Fig. 9, sampling variability, statistical errors (false positives, false negatives, inflation and sign errors), and out-of-sample multivariate associations ($r_{pred}$, $r_{CV1}$) were plotted as a function of sample size (y-axis: 0–1 for sampling variability ($r$), 0–100% for statistical errors (cumulative sum across all four error types), 0–100% for out-of-sample associations). To account for differences between in-sample and out-of-sample multivariate associations, out-of-sample multivariate associations were normalized by the mean in-sample (discovery) correlation at the full sample size. All three curves (sampling variability, statistical errors, and out-of-sample association) were based on the largest univariate and multivariate brain-wide association (RSFC with cognitive ability).

### Reporting summary
Further information on research design is available in the Nature Research Reporting Summary linked to this paper.

## Data availability
Participant level data from all datasets (ABCD, HCP, UKB) are openly available pursuant to individual consortium-level data access rules. The ABCD data repository grows and changes over time (https://nda.nih.gov/abcd). The ABCD data used in this report came from ABCD collection 3165 and the Annual Release 2.0 (https://doi.org/10.15154/1503209). The UK Biobank is a large-scale biomedical database and research resource containing genetic, lifestyle and health information from half a million UK participants (www.ukbiobank.ac.uk). UK Biobank's database, which includes blood samples, heart and brain scans and genetic data of the 500,000 volunteer participants, is globally accessible to approved researchers who are undertaking health-related research that is in the public interest. Data were provided, in part, by the Human Connectome Project, WU-Minn Consortium (principal investigators: D. Van Essen and K. Ugurbil; 1U54MH091657) funded by the 16 NIH institutes and centers that support the NIH Blueprint for Neuroscience Research; and by the McDonnell Center for Systems Neuroscience at Washington University. Some data used in the present study are available for download from the Human Connectome Project (www.humanconnectome.org). Users must agree to data use terms for the HCP before being allowed access to the data and ConnectomeDB; details are provided at https://www.humanconnectome.org/study/hcp-young-adult/data-use-terms. Source data are provided with this paper.

## Code availability
Analysis code specific to this study can be found at https://gitlab.com/DosenbachGreene/bwas. Code for processing ABCD and UKB data can be found at https://github.com/DCAN-Labs/abcd-hcp-pipeline. MRI

data analysis code can be found at https://github.com/ABCD-STUDY/nda-abcd-collection-3165. FIRMM software is available at https://firmm.readthedocs.io/en/latest/release_notes/ (the ABCD Study used version 3.0.14). The MuMln R package (version 1.43.17) is available at https://cran.r-project.org/web/packages/MuMIn/index.html.

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

**Acknowledgements** We thank A. M. Dale, T. L. Jernigan, W. Zhao, C. Makowski, C. C. Fan and C. Palmer for their thoughtful comments on the manuscript. This work was supported by NIH grants MH100019 (S.M., N.A.S., A.M. and T.O.L.,), MH121518 (S.M.), DA007261 (D.F.M.), NS090978 (B.P.K.), DA007261 (A.S.H.), MH125023 (M.R.D), NS110332 (D.J.N.), NS115672 (A.Z.), MH112473 (T.O.L.), MH104592 (D.J.G.), AA02969 (S.M.M.), DA041148 (D.A.F.), DA04112 (D.A.F.), MH115357 (D.A.F.), MH096773 (D.A.F. and N.U.F.D.), MH122066 (D.A.F. and N.U.F.D.), MH121276 (D.A.F. and N.U.F.D.), MH124567 (D.A.F. and N.U.F.D.), NS088590 (N.U.F.D.), and the Andrew Mellon Predoctoral Fellowship (B.T.-C.), Lynne and Andrew Redleaf Foundation (D.A.F.), Kiwanis Neuroscience Research Foundation (N.U.F.D.) and the Jacobs Foundation grant 2016121703 (N.U.F.D.). ABCD Study: data used in the preparation of this article, in part, were obtained from the ABCD Study (https://abcdstudy.org), held in the NIMH Data Archive (NDA). This is a multisite, longitudinal study designed to recruit more than 10,000 children aged 9–10 and follow them over 10 years into early adulthood. The ABCD Study is supported by the National Institutes of Health and additional federal partners under award numbers U01DA041022, U01DA041028, U01DA041048, U01DA041089, U01DA041106, U01DA041117, U01DA041120, U01DA041134, U01DA041148, U01DA041156, U01DA041174, U24DA041123, U24DA041147, U01DA041093 and U01DA041025. A full list of supporters is available at https://abcdstudy.org/federal-partners.html. A listing of participating sites and a complete listing of the study investigators can be found at https://abcdstudy.org/scientists/workgroups/. ABCD consortium investigators designed and implemented the study and/or provided data but did not necessarily participate in analysis or writing of this report. This manuscript reflects the views of the authors and may not reflect the opinions or views of the NIH or ABCD consortium investigators. HCP Study: data were provided, in part, by the Human Connectome Project, WU-Minn Consortium (U54 MH091657) funded by the 16 NIH Institutes and Centers that support the NIH Blueprint for Neuroscience Research; and by the McDonnell Center for Systems Neuroscience at Washington University. UK Biobank study: this research has been conducted, in part, using data from UK Biobank (www.ukbiobank.ac.uk). UK Biobank is generously supported by its founding funders the Wellcome Trust and UK Medical Research Council, as well as the Department of Health, Scottish Government, the Northwest Regional Development Agency, British Heart Foundation and Cancer Research UK. XSEDE and Pittsburgh Supercomputing Center: this work used the Extreme Science and Engineering Discovery Environment (XSEDE), which is supported by National Science Foundation grant number ACI-1548562. Specifically, it used the Bridges system, which is supported by NSF award number ACI-1445606, at the Pittsburgh Supercomputing Center (PSC TG-IBN200009). MIDB, NGDR and MSI: this work used the storage and computational resources provided by the Masonic Institute for the Developing Brain (MIDB) the Neuroimaging Genomics Data Resource (NGDR) and the Minnesota Supercomputing Institute (MSI). NGDR is supported by the University of Minnesota Informatics Institute through the MnDRIVE initiative in coordination with the College of Liberal Arts, Medical School and College of Education and Human Development at the University of Minnesota. Daenerys NCCR: this work used the storage and computational resources provided by the Daenerys Neuroimaging Community Computing Resource (NCCR). The Daenerys NCCR is supported by the McDonnell Center for Systems Neuroscience at Washington University, the Intellectual and Developmental Disabilities Research Center (IDDRC; P50 HD103525) at Washington University School of Medicine and the Institute of Clinical and Translational Sciences (ICTS; UL1 TR002345) at Washington University School of Medicine.

**Author contributions** Conception: S.M., B.T.-C., D.A.F. and N.U.F.D. Design: S.M., B.T.-C., F.J.C., D.F.M., B.T.T.Y., B.L., D.A.F. and N.U.F.D. Data acquisition, analysis and interpretation: S.M., B.T.-C., F.J.C., D.F.M., B.P.K., A.S.H., M.R.D., W.F., R.L.M., T.J.H., S.M.M., S.K., E.F., O.M.-D., A.M.G., E.A.E., A.J.P., M.C., O.D., L.A.M., G.M.C., J.U., K.S., B.J.L., J.C.W., T.P., T.O.L., D.J.G., S.E.P., H.G., W.K.T., T.E.N., B.T.T.Y., D.M.B., B.L., D.A.F. and N.U.F.D. Manuscript writing, revising: S.M., B.T.-C., F.J.C., D.F.M., B.P.K., A.S.H., M.R.D., W.F., R.L.M., T.J.H., E.F., O.M.-D., A.M.G., E.A.E., A.J.P., M.C., O.D., L.A.M., G.M.C., J.U., K.S., A.T., J.C., D.J.N., A.Z., N.A.S., A.N.V., A.M., R.J.C., T.O.L., D.J.G., S.E.P., H.G., W.K.T., T.E.N., B.T.T.Y., D.M.B., B.L., D.A.F. and N.U.F.D.

**Competing interests** E.A.E., D.A.F and N.U.F.D. have a financial interest in NOUS Imaging Inc. and may financially benefit if the company is successful in marketing FIRMM motion-monitoring software products. O.M.-D., E.A.E., A.N.V., D.A.F. and N.U.F.D. may receive royalty income based on FIRMM technology developed at Washington University School of Medicine and Oregon Health and Sciences University and licensed to NOUS Imaging Inc. D.A.F. and N.U.F.D. are co-founders of NOUS Imaging Inc. and E.A.E. is a former employee of NOUS Imaging. These potential conflicts of interest have been reviewed and are managed by Washington University School of Medicine, Oregon Health and Sciences University and the University of Minnesota. The other authors declare no competing interests.

**Additional information**
**Correspondence and requests for materials** should be addressed to Scott Marek, Brenden Tervo-Clemmens, Damien A. Fair or Nico U. F. Dosenbach.

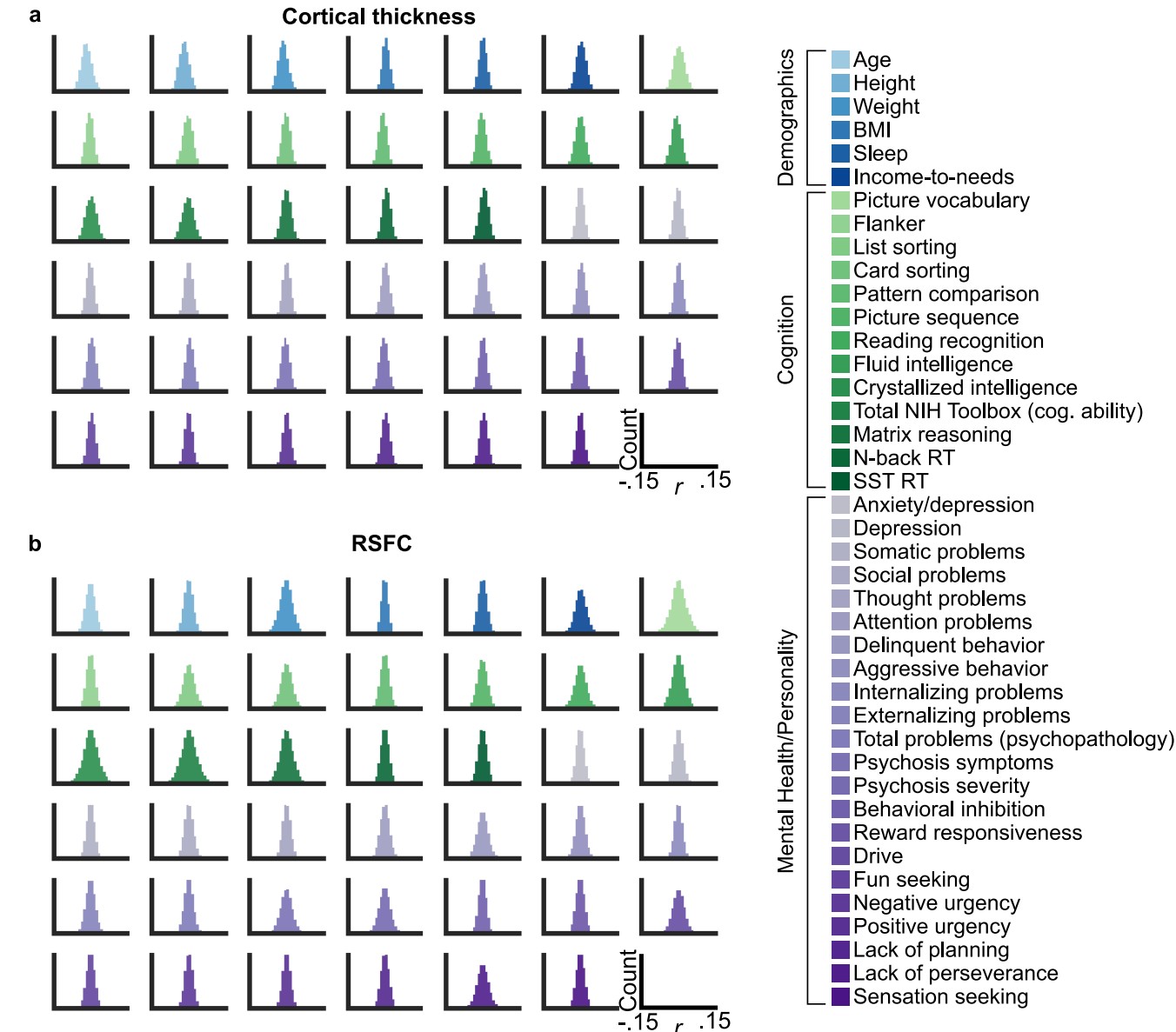

**a** Cortical thickness

**b** RSFC

**Extended Data Fig. 1 | Distributions of brain-wide association effect sizes by imaging modality and behavioral phenotype.** Histograms of all **(a)** cortical thickness and **(b)** resting-state functional connectivity (RSFC) associations, with demographic, cognitive, and mental health/personality variables. Correlations (*r*; linear bivariate) between brain measures and behavioral phenotypes were computed at multiple levels of scale (cortical thickness: vertices, regions of interest (ROIs), networks; RSFC: ROI-ROI pairs (edges), principal components, networks). The ordering of subgraphs follows the ordering of measures in the legend. All data shown are from the ABCD Study (*n* = 3,928).

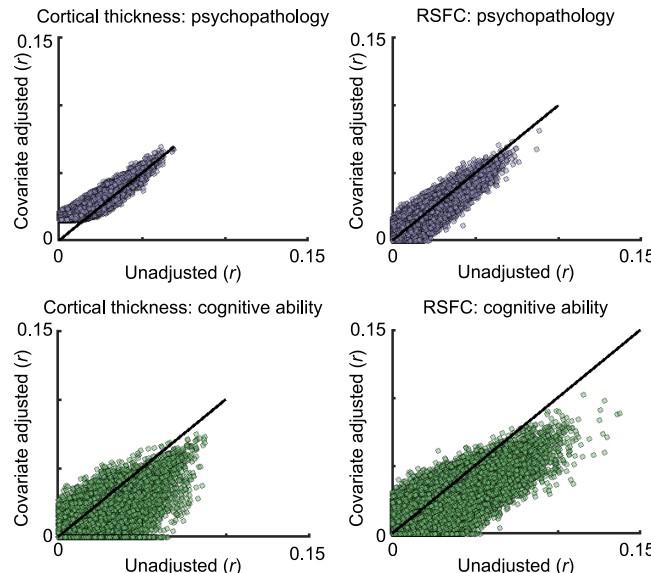

**Extended Data Fig. 2 | Impact of sociodemographic covariates on brain-wide association effect sizes.** The influence of sociodemographic covariates (race, gender, parental marital status, parental income, hispanic versus non-hispanic ethnicity, family, data collection site) on BWAS (brain-wide association studies) effect sizes was examined in the ABCD Study dataset ($n$ = 3,587 with complete cases for this analysis) through the model comparison strategy developed by the ABCD Data Analysis and Informatics Core and used in the Data Exploration and Analysis Portal (deap.nimhda.org). The percentages of variance explained by fixed effects in multilevel models (pseudo-$R^2$) were calculated with the MuMIn package in R (1.43.17) and square root transformed to approximate an absolute-value BWAS correlation ($|r|$). The estimated BWAS effect sizes ($|r|$) prior to covariate adjustment are plotted on the x-axis and those after sociodemographic covariate adjustment on the y-axis. Values below the identity line indicate a reduction in effect size after covariate adjustment, values above an increase in effect size. BWAS models with and without covariate adjustment always included cognitive ability or psychopathology as the outcome variable and nested random effects of family and data collection site, in order to maximize comparability for subsequent fixed effects model comparisons. BWAS effect sizes without covariate adjustment were taken from models that only included these random effects, the brain feature of interest (cortical thickness [vertex]/RSFC [edge]) as a single fixed effect, and the psychological phenotype (cognitive ability/ psychopathology). BWAS effect sizes without covariate adjustment estimated the unique, covariate-adjusted effect linking the brain feature of interest to the psychological phenotype by comparing a model with sociodemographic fixed effects but no brain feature fixed effect, to one with both the sociodemographic fixed effects and the brain feature. The difference in pseudo-$R^2$ (subsequently transformed to $|r|$) represents the additional fixed-effect variance the brain feature explained beyond the sociodemographic covariates.

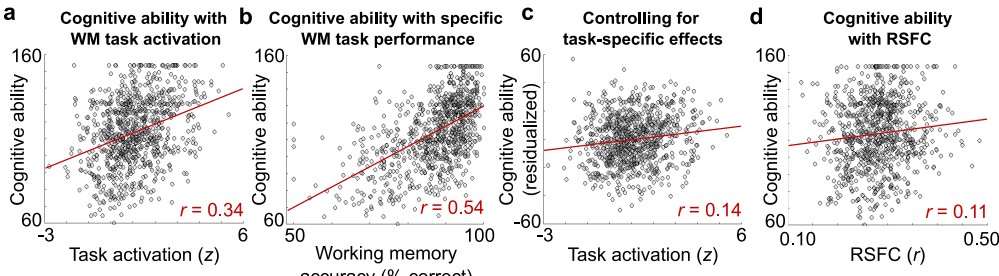

**Extended Data Fig. 3 | Brain-wide association effect sizes derived from functional MRI (fMRI) task activations are similar to resting-state functional connectivity (RSFC). (a)** Cognitive ability (NIH Toolbox total composite score) plotted as a function of dorsal attention network working memory task activation (z). Note that this correlation with fMRI task activation (r = 0.34) is much larger than the largest replicated univariate effect size for RSFC. **(b)** Cognitive ability plotted as a function of working memory task accuracy. Individual differences in cognitive ability (phenotype of interest) are strongly correlated with individual differences in working memory (r = 0.54).

Thus, task-specific effects (behavioral performance) confound links between brain function and the phenotype of interest (e.g. cognitive ability). **(c)** Residualizing the behavioral phenotype of interest (cognitive ability) with respect to individual differences in working memory task accuracy (task-specific effect) produces an association between task fMRI and cognitive ability (r = 0.14) similar to the **(d)** the association between dorsal attention network RSFC and cognitive ability (r = 0.11). Data shown are from the HCP Study (n = 844).

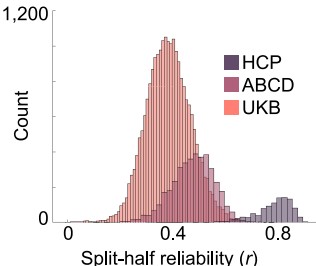

**Extended Data Fig. 4 | Split-half reliability of resting-state functional connectivity (RSFC) in HCP, ABCD and UKB study samples.** Distribution of within-person split-half reliability[33] of ROI (333 cortical ROIs from Gordon et al.[61]) connectivity matrices derived from RSFC data. The UKB data contain a single 6 min. resting-state run; the ABCD Study collected 4 x 5 min. runs (20 min. total), and the HCP collected 4 x 15 min. runs of resting-state data (60 min. total).

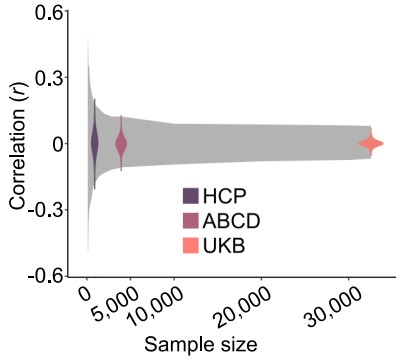

**Extended Data Fig. 5 | Effect size distributions for HCP, ABCD, UKB studies and expected sampling variability.** To determine whether smaller effect sizes in larger samples can be explained by the expected reduction of sampling variability, we estimated sampling variability (grey) for the full range of BWAS (brain-wide association studies) effect sizes observed in UKB (edge-wise resting-state functional connectivity [RSFC]; cognitive ability) as a function of sample size (x-axis). As in our primary ABCD analyses, UKB effects were resampled using a bootstrap procedure (1,000 iterations per edge). The actual distributions of the HCP, ABCD, and UKB BWAS effect sizes were then visualized relative to the expected sampling variability in UKB across sample sizes (grey). Consistent with an inflation of BWAS effect sizes due to sampling variability, relatively larger BWAS effect sizes in HCP ($n$ = 900) and ABCD ($n$ = 3,928) align with effect sizes in subsamples of the UKB data at corresponding sample sizes.

**a** **ABCD vs. HCP sampling variability**

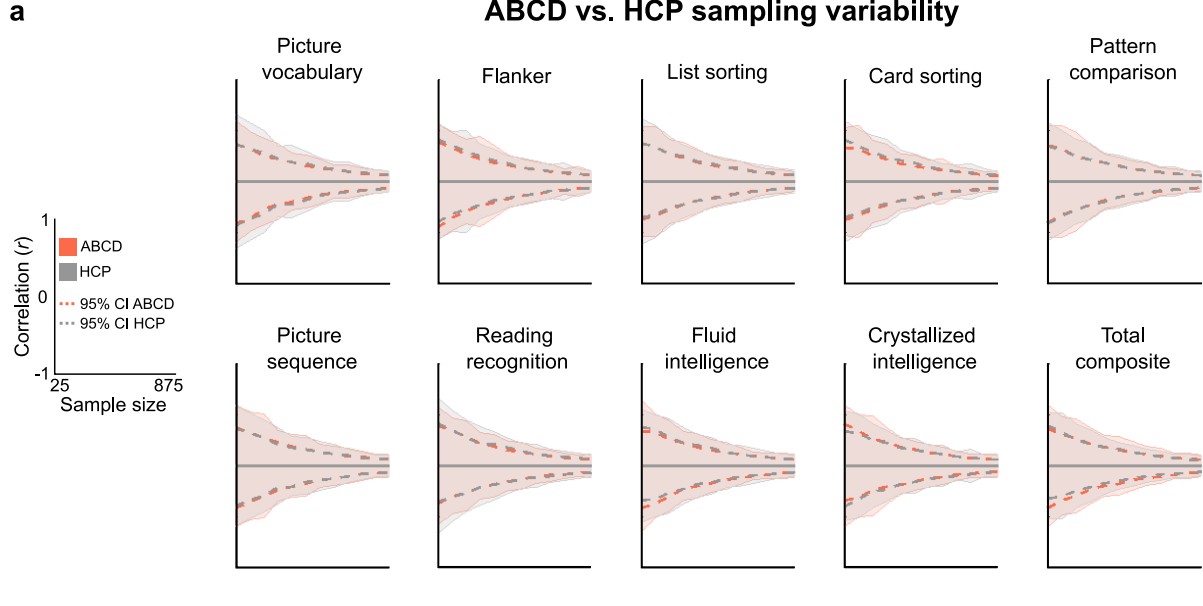

**b** **Single site vs. multi site sampling variability**

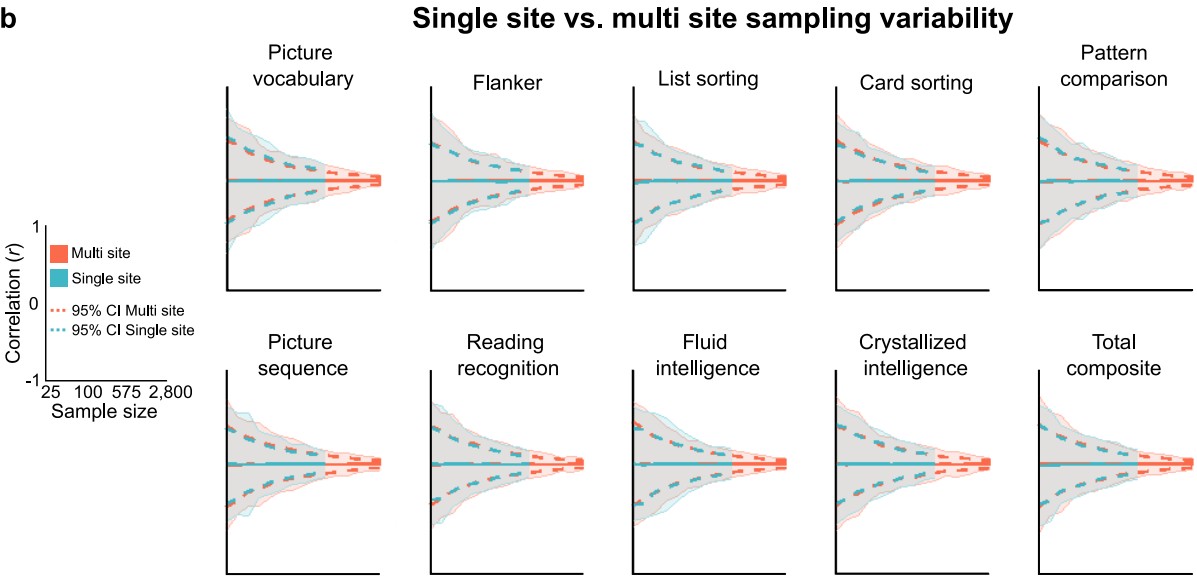

**Extended Data Fig. 6 | Comparison of single- and multi-site BWAS (brain-wide association studies) sampling variability. (a)** Sampling variability of resting-state functional connectivity (RSFC) associations with the NIH Toolbox subscales in equally-sized samples (*n* = 877) from HCP (grey) and ABCD (red). Effect sizes (center of error bands) were matched across datasets (*r* = 0.06) to isolate sampling variability for a given effect. **(b)** Sampling variability of RSFC associations with the NIH Toolbox subscales in a single-site ABCD sample (site 16; *n* = 603; teal) and every other ABCD site (*n* = 3,325; red). Effect sizes (center of error bands) were matched across datasets (*r* = 0.06).

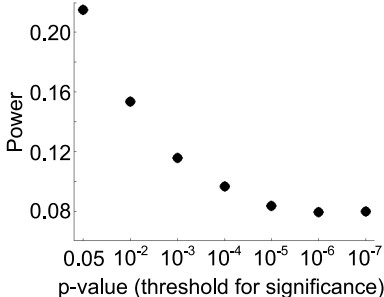

**Extended Data Fig. 7 | Relationship between power and statistical threshold.** Statistical power (1 − false negative rate) is plotted as a function of the $P$ value (two-tailed; $< 0.05$, $< 10^{-2}$, $< 10^{-3}$, $< 10^{-4}$, $< 10^{-5}$, $< 10^{-6}$, $< 10^{-7}$) used for significance testing in the denoised ABCD Study sample ($n = 3{,}928$). $P < 0.05$ represents an uncorrected threshold, whereas $P < 10^{-7}$ represents a Bonferroni correction. More stringent control for multiple comparisons decreases power and increases sample size requirements.

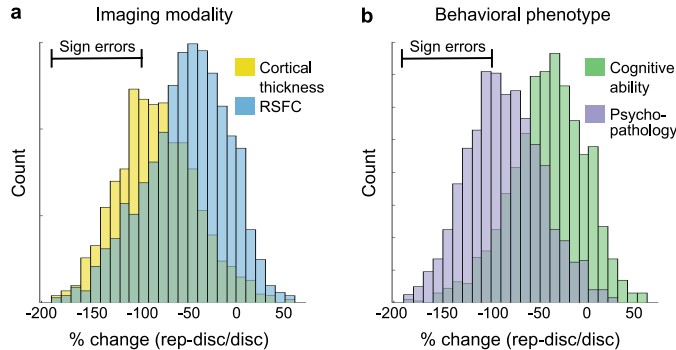

**Extended Data Fig. 8 | Inflation of univariate BWAS (brain-wide association studies) effect sizes (top 1% largest) by imaging modality and behavioral phenotype.** Better out-of-sample replication is indexed by a smaller difference between the discovery and replication datasets effect sizes (right side of histogram). Negative values indicate that an association was inflated in the discovery dataset, relative to what was observed in the replication dataset. Out-of-sample reductions in effect sizes greater than 100% reflect sign errors. The leftward shift of cortical thickness relative to resting-state functional connectivity (RSFC), and for psychopathology relative to cognitive ability indicates worse univariate BWAS reproducibility.

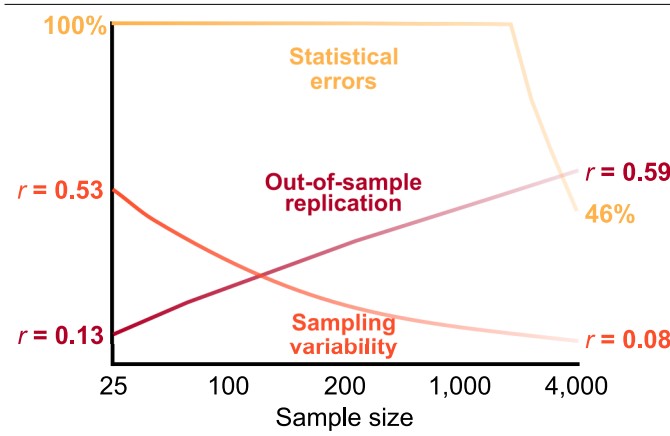

**Extended Data Fig. 9 | Influence of sample size on the robustness of brain-wide associations.** Trajectories of sampling variability (99% confidence interval; orange), statistical error rates (cumulative sum of false negatives, false positives, magnitude errors, sign errors; yellow), and support vector regression (SVR) out-of-sample association strength (as % of full in-sample association; dark red) as a function of sample size. Sample size (*n* ~ 4,000) represents a full sample (discovery + replication datasets of ~2,000 each). Data shown are from ABCD Study.

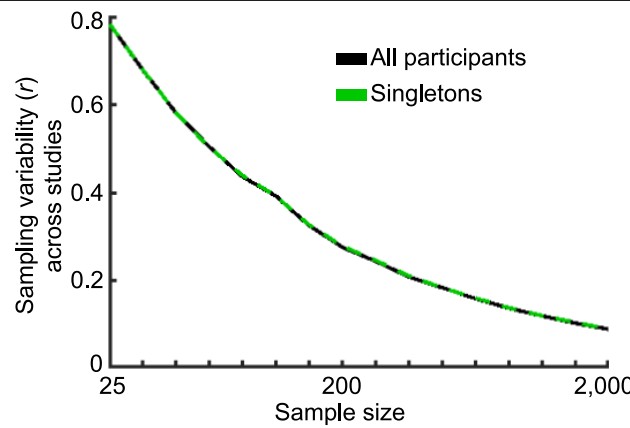

**Extended Data Fig. 10 | Sampling variability is nearly identical when considering singletons vs. all participants.** Data were from the ABCD Study sample. Sampling variability (y-axis) as a function of sample size (x-axis; $n = 25$, 35, 45, 60, 80, 100, 145, 200, 256, 350, 460, 615, 825, 1,100, 1,475, 2,000) for all participants (black) and singletons only (twins and siblings excluded; green). Sampling variability was quantified as the difference between the upper and lower 95% confidence interval across 1,000 bootstraps (resampled with replacement) across all 77,421 resting-state functional connectivity (RSFC; edges) associations with cognitive ability. The effect size magnitudes were likewise nearly identical in size-matched resamples (singletons-only [$n = 2,528$]: median $|r| = 0.017$; siblings-included [$n = 2,528$]: median $|r| = 0.020$).

# Reporting Summary

## Statistics

For all statistical analyses, confirm that the following items are present in the figure legend, table legend, main text, or Methods section.

| n/a | Confirmed | |
|---|---|---|
| ☐ | ☒ | The exact sample size (*n*) for each experimental group/condition, given as a discrete number and unit of measurement |
| ☐ | ☒ | A statement on whether measurements were taken from distinct samples or whether the same sample was measured repeatedly |
| ☐ | ☒ | The statistical test(s) used AND whether they are one- or two-sided *Only common tests should be described solely by name; describe more complex techniques in the Methods section.* |
| ☐ | ☒ | A description of all covariates tested |
| ☐ | ☒ | A description of any assumptions or corrections, such as tests of normality and adjustment for multiple comparisons |
| ☐ | ☒ | A full description of the statistical parameters including central tendency (e.g. means) or other basic estimates (e.g. regression coefficient) AND variation (e.g. standard deviation) or associated estimates of uncertainty (e.g. confidence intervals) |
| ☐ | ☒ | For null hypothesis testing, the test statistic (e.g. *F*, *t*, *r*) with confidence intervals, effect sizes, degrees of freedom and *P* value noted *Give P values as exact values whenever suitable.* |
| ☒ | ☐ | For Bayesian analysis, information on the choice of priors and Markov chain Monte Carlo settings |
| ☒ | ☐ | For hierarchical and complex designs, identification of the appropriate level for tests and full reporting of outcomes |
| ☐ | ☒ | Estimates of effect sizes (e.g. Cohen's *d*, Pearson's *r*), indicating how they were calculated |

*Our web collection on statistics for biologists contains articles on many of the points above.*

## Software and code

Policy information about availability of computer code

| Data collection | No software was used for data collection. Neuroimaging and behavioral data were from existing, open-source datasets (ABCD, UKB, HCP) whose acquisition's are presented in detail in previous work. The ABCD Study data were collected between 2016-2018. The HCP data were collected between 2010-2016. The UKB data collection started in 2014 and 2015 to demonstrate the feasibility of high-throughput imaging and to finalize the imaging protocols required for the main phase. Funding was then released to extend the imaging enhancement to an additional 95,000 participants, with data collection estimated to finish by 2023. |
|---|---|
| Data analysis | MRI data analysis code can be found here: https://github.com/ABCD-STUDY/nda-abcd-collection-3165 ABCD and UKB MRI data processing code can be found here https://github.com/DCAN-Labs/abcd-hcp-pipeline Manuscript analysis code can be found here https://gitlab.com/DosenbachGreene/bwas FIRMM software: https://firmm.readthedocs.io/en/latest/release_notes/. ABCD uses version 3.0.14. MuMln R package: https://cran.r-project.org/web/packages/MuMln/index.html. Version 1.43.17. |

For manuscripts utilizing custom algorithms or software that are central to the research but not yet described in published literature, software must be made available to editors and reviewers. We strongly encourage code deposition in a community repository (e.g. GitHub). See the Nature Portfolio guidelines for submitting code & software for further information.

## Data

Policy information about availability of data

All manuscripts must include a data availability statement. This statement should provide the following information, where applicable:

- Accession codes, unique identifiers, or web links for publicly available datasets
- A description of any restrictions on data availability
- For clinical datasets or third party data, please ensure that the statement adheres to our policy

Participant level data from all datasets (ABCD, HCP, UKB) is openly available pursuant to individual, consortia-level data access rules. The ABCD data repository grows and changes over time. The ABCD data used in this report came from ABCD collection 3165 and the Annual Release 2.0, DOI 10.15154/1503209.
The UK Biobank is a large-scale biomedical database and research resource containing genetic, lifestyle and health information from half a million UK participants (www.ukbiobank.ac.uk). UK Biobank's database, which includes blood samples, heart and brain scans and genetic data of the 500,000 volunteer participants, is globally accessible to approved researchers who are undertaking health-related research that's in the public interest.
Data were provided, in part, by the Human Connectome Project, WU-Minn Consortium (Principal Investigators: David Van Essen and Kamil Ugurbil; 1U54MH091657) funded by the 16 NIH Institutes and Centers that support the NIH Blueprint for Neuroscience Research; and by the McDonnell Center for Systems Neuroscience at Washington University. Some data used in the present study are available for download from the Human Connectome Project (www.humanconnectome.org). Users must agree to data use terms for the HCP before being allowed access to the data and ConnectomeDB, details are provided at https://www.humanconnectome.org/study/hcp-young-adult/data-use-terms.
No new data were collected for this manuscript. Across the ABCD, HCP, and UKB, we downloaded data between 01/2019 - 10/2021. We did not use any specific software for downloading the data. For details on data collection in ABCD (baseline data), see Casey et al., 2018; in HCP (1200 release) see Van Essen et al., 2013; in UKB see Sudlow et al., 2015.

# Field-specific reporting

Please select the one below that is the best fit for your research. If you are not sure, read the appropriate sections before making your selection.

☐ Life sciences   ☒ Behavioural & social sciences   ☐ Ecological, evolutionary & environmental sciences

For a reference copy of the document with all sections, see nature.com/documents/nr-reporting-summary-flat.pdf

# Behavioural & social sciences study design

All studies must disclose on these points even when the disclosure is negative.

| | |
|---|---|
| Study description | Quantitative analyses of the magnitude and reproducibility of cross-sectional associations between neuroimaging measures and psychological/psychiatric phenotypes. |
| Research sample | Because our main focus was to estimate the effect size of BWAS, which requires a very large sample, this project uses open-access data from three of the largest community-recruited neuroimaging samples (ABCD, HCP, UKB) that contain both structural and functional MRI data. Combined, the aggregate sample is one of the largest (N~50,000) and most representative neuroimaging samples, although none of the samples is fully population representative. See manuscript for extended discussion. |
| Sampling strategy | All samples were recruited from the community (ABCD & HCP from the USA and UK Biobank from the United Kingdom). Individual samples (ABCD, HCP, UKB) used unique sample size calculations and sampling strategies which are discussed in prior work with these open source datasets (Casey et al., 2018, Van Essen et al., 2013, and Littlejohns et al., 2020, respectively). |
| Data collection | All data were from existing data repositories and were downloaded between 01/2019 - 10/2021. Data used in the manuscript were from existing large consortia datasets (ABCD: see Casey et al., 2018 & Barch et al., 2018; HCP: We used data from the 1200 subjects data release (van Essen et al., 2013); The UKB brain imaging component has been described in Miller et al. (2016), and the processing and quality control described in Alfaro-Almagro et al. (2018)). Because we did not personally collect any of the data used in this manuscript, all data were from existing data repositories and researchers were therefore not blind to the source of the data. |
| Timing | ABCD: see Casey et al., 2018<br>HCP: see van Essen et al., 2013<br>UKB: see Miller et al., 2016 |
| Data exclusions | In ABCD, we used strict inclusion criteria with regard to head motion. Specifically, inclusion criteria for the current project (see ref 30 in manuscript for broader ABCD inclusion criteria) consisted of at least 600 frames (8 minutes) of low-motion (filtered FD<0.08) resting state functional connectivity data. Our final dataset consisted of data from a total of N=3,928 youth across the discovery (N=1,964) and replication (N=1,964) sets. The final discovery and replication sets did not differ in mean FD (ΔM=0.002 , t=0.60, p=0.55) or total frames included (ΔM=6.4 , t=0.94, p=0.35). The subject lists for ARMS samples and our associated matrices will be released in the ABCD-BIDS Community Collection (ABCD collection 3165) for community use.<br>For HCP data, we used similar data quantity inclusion, as well as an FD < 0.20 (unfiltered FD). This resulted in the inclusion of N=900 individuals (N=877 across all NIH Toolbox subscales).<br>Given the UK Biobank data only contains 6 mins of resting state data, we did not exclude any subjects due to low quantities of data post motion censoring (FD < 0.08, filtered). |
| Non-participation | N/A |

| Randomization | All three samples were observational studies and no randomization was used. |
|---|---|

# Reporting for specific materials, systems and methods

We require information from authors about some types of materials, experimental systems and methods used in many studies. Here, indicate whether each material, system or method listed is relevant to your study. If you are not sure if a list item applies to your research, read the appropriate section before selecting a response.

## Materials & experimental systems

| n/a | Involved in the study |
|---|---|
| ☒ | ☐ Antibodies |
| ☒ | ☐ Eukaryotic cell lines |
| ☒ | ☐ Palaeontology and archaeology |
| ☒ | ☐ Animals and other organisms |
| ☐ | ☒ Human research participants |
| ☒ | ☐ Clinical data |
| ☒ | ☐ Dual use research of concern |

## Methods

| n/a | Involved in the study |
|---|---|
| ☒ | ☐ ChIP-seq |
| ☒ | ☐ Flow cytometry |
| ☐ | ☒ MRI-based neuroimaging |

## Human research participants

Policy information about studies involving human research participants

| Population characteristics | See above. |
|---|---|
| Recruitment | *Describe how participants were recruited. Outline any potential self-selection bias or other biases that may be present and how these are likely to impact results.* |
| Ethics oversight | The ABCD Study obtained centralized institutional review board approval from the University of California, San Diego, and each of the 21 study sites obtained local institutional review board approval. Ethical regulations were followed during data collection and analysis. Parents or caregivers provided written informed consent, and children gave written assent. |

Note that full information on the approval of the study protocol must also be provided in the manuscript.

## Magnetic resonance imaging

### Experimental design

| Design type | resting-state fMRI, task-based fMRI; structural (cortical thickness) MRI |
|---|---|
| Design specifications | ABCD resting state: 4, 5 min runs, eyes open<br>HCP resting state: 4, 15 min runs, eyes open<br>UKB resting state: 1, 6 min run, eyes open |
| Behavioral performance measures | Primary analyses use cognitive assessments from the NIH Toolbox and psychopathology assessment Child Behavior Checklist (see manuscript for individual subscales, total of 41 ) included in standard data releases and discussed in detail perviously (Barch et al., 2018) |

### Acquisition

| Imaging type(s) | Resting-state fMRI, task-fMRI, structural (cortical thickness) MRI |
|---|---|
| Field strength | 3 Tesla |
| Sequence & imaging parameters | Primary analyses use open-source distributed fMRI and MR data that adhere to consortia guidelines (see Casey et al., 2018, Van Essen et al., 2013, and Miller et al., 2016 for ABCD, HCP, and UKB parameters, respectively). |
| Area of acquisition | Whole brain |
| Diffusion MRI | ☐ Used   ☒ Not used |

### Preprocessing

| Preprocessing software | Preprocessing of ABCD was done using a suite of tools. All code can be found here: https://github.com/ABCD-STUDY/nda-abcd-collection-3165. Individual datasets  (ABCD, UKB, HCP) and individual study sites (e.g., ABCD site 1 versus site 2) used unique sequence and imaging parameters which are discussed in prior work introducing these open-source datasets. |

| Normalization | 1) PreFreesurfer normalizes anatomical data. This normalization entails brain extraction, denoising, and then bias field correction on anatomical T1 and/or T2 weighted data. The ABCD-HCP pipeline includes two additional modifications to improve output image quality. ANTs 65 DenoiseImage models scanner noise as a Rician distribution and attempts to remove such noise from the T1 and T2 anatomical images. Additionally, ANTs N4BiasFieldCorrection attempts to smooth relative image histograms in different parts of the brain and improves bias field correction. 2) FreeSurfer 1 constructs cortical surfaces from the normalized anatomical data. This stage performs anatomical segmentation, white/grey and grey/CSF cortical surface construction, and surface registration to a standard surface template. Surfaces are refined using the T2 weighted anatomical data. Mid-thickness surfaces, which represent the average of white/grey and grey/CSF surfaces, are generated here. 3) PostFreesurfer converts prior outputs into an HCP-compatible format (i.e. CIFTIs) and transforms the volumes to a standard volume template space using ANTs nonlinear registration, and the surfaces to the standard surface space via spherical registration. |
|---|---|
| Normalization template | The "Vol" stage corrects for functional distortions via reverse-phase encoding spin-echo images. All resting state runs underwent intensity normalization to a whole brain mode value of 1000, within run correction for head movement, and functional data registration to the standard template (MNI). Atlas transformation was computed by registering the mean intensity image from each BOLD session to the high resolution T1 image, and then applying the anatomical registration to the BOLD image. This atlas transformation, mean field distortion correction, and resampling to 3-mm isotropic atlas space were combined into a single interpolation using FSL's 66 applywarp tool. The "Surf" stage projects the normalized functional data onto the template surfaces. |
| Noise and artifact removal | Additional BOLD preprocessing steps were executed to reduce spurious variance unlikely to reflect neuronal activity 46. First, a respiratory filter was used to improve FD estimates calculated in the volume ("vol") stage68. Second, temporal masks were created to flag motion-contaminated frames using the improved FD estimates 63. Frames with a filtered FD>0.3mm were flagged as motion-contaminated for nuisance regression only. After computing the temporal masks for high motion frame censoring, the data were processed with the following steps: (i) demeaning and detrending, (ii) interpolation across censored frames using least squares spectral estimation of the values at censored frames so that continuous data can be (iii) denoised via a GLM with whole brain, ventricular, and white matter signal regressors, as well as their derivatives. Denoised data were then passed through (iv) a band-pass filter (0.008 Hz<f<0.10 Hz) without re-introducing nuisance signals 69 or contaminating frames near high motion frames. |
| Volume censoring | Yes, ABCD data were censored at a filtered frame-wise displacement of < 0.08mm and HCP data were filtered using a non-filtered framewise displacement of <0.20mm. |

## Statistical modeling & inference

| Model type and settings | Mass univariate and multivariate (support vector regression, canonical correlation analysis). Multiple parameterizations of each of these models were explored with the stated goal being to determine field-wide reproducibility in brain-phenotype association studies (see manuscript). |
|---|---|
| Effect(s) tested | As the primary aim of the paper was to determine the general reproducibility of brain-phenotype effects, multiple scales and combinations of effects were examined. Owing to the cross-sectional, nature of these studies, all effects are between-person associations. |

Specify type of analysis: ☐ Whole brain  ☐ ROI-based  ☒ Both

| Anatomical location(s) | Parcel-level and network-level analyses utilized the field-standard Gordon et al., 2016, Cerebral Cortex, and Seitzman et al., 2020, NeuroImage. Vertex-wise and voxel-wise data were extracted from Ciftis. |
|---|---|

| Statistic type for inference (See Eklund et al. 2016) | Multiple levels of neuroanatomical scale were used, including voxels, regions of interest, and networks. |
|---|---|
| Correction | As the primary aim of the paper was to determine the general reproducibility of brain-phenotype effects, multiple levels of significance values and correction were used, ranging from uncorrected to bonferroni (FWER) correction. |

## Models & analysis

| n/a | Involved in the study |
|---|---|
| ☐ | ☒ Functional and/or effective connectivity |
| ☒ | ☐ Graph analysis |
| ☐ | ☒ Multivariate modeling or predictive analysis |

| Functional and/or effective connectivity | Pearson correlation |
|---|---|
| Multivariate modeling and predictive analysis | Multivariate Out-of-Sample Replication: Support Vector Regression (SVR) Support vector regression (SVR) with a linear kernel was performed using the e1071 package in the R environment (version 3.5.2) to predict primary phenotypes (psychopathology, cognitive ability) and other demographics and psychological phenotypes from individual differences in either RSFC or cortical thickness. One hundred bootstrap samples (sampling with replacement) were generated for each sample size. Hyperparameter tuning was examined in 1) split halves of the full discovery sample for multiple cognitive (NIH Toolbox) and psychopathology (CBCL) symptoms and 2) 10-fold cross-validation within the full discovery sample for primary phenotypes (psychopathology, cognitive ability), but did not appreciably change out-of-sample prediction estimates to the replication sample (e.g., average out-of-sample correlation difference between tuned and non-tuned models : RSFC = -0.006, Cortical Thickness = 0.014). Fig. 4A,C use default |

hyperparameters and PCA dimensionality reduction (with a threshold of 50% variance explained in the discovery set, for each sample size) prior to SVR, given that this procedure balanced out-of-sample prediction and model complexity for nearly all model types. Replication set data were not used to estimate principal components, but rather replication set data were projected into component space via independently estimated loading matrices for each subsample of the discovery set to prevent bias. An alternative strategy of univariate feature ranking was also examined, where SVR models were trained on the 5,000, 10,000, or 15,000 vertices (cortical thickness) or edges (RSFC) with the highest correlation to the variable of interest in the training dataset, but this approach resulted in lower out-of-sample prediction. Significance thresholds for out-of-sample replication were estimated via permutation testing (1,000 iterations) with models trained on the full discovery set (RSFC: N=1964; cortical thickness: N=1,814) and tested on the full replication set.

Multivariate Out-of-Sample Replication: Canonical Correlation Analysis (CCA)
Canonical correlation analysis (CCA) was performed using Matlab's (2019b) cannoncor.m function to predict the NIH Toolbox and CBCL from individual differences in either RSFC or cortical thickness. Equivalent bootstrapping and subsampling of the discovery set were tested and applied to the replication set, as in the SVR analyses. In order to model sampling variability across sample sizes, 100 bootstrap (sampling with replacement) samples were generated for each sample size. As with SVR, Fig. 4B,D used principal-component analysis (PCA) dimensionality reduction (threshold of 20% variance explained in the discovery set, for each sample size) prior to CCA given that this maximised out-of-sample prediction. CCA models were fit on iteratively larger subsamples of the discovery (in-sample) data set. The first canonical vector was extracted and applied to the full replication (out-of-sample) brain data to predict replication set (out-of-sample) behavioral phenotype. Prediction accuracy was quantified by expressing the correlation between the matrix products of the first canonical vector (from the discovery set) and replication brain and phenotypic data. Significance thresholds for out-of-sample replication were estimated via permutation testing (1,000 iterations) with models trained on the full discovery set (RSFC: N=1964; cortical thickness: N=1,814) and tested on the full replication set.

