## [Peer Review File · Nature]

Manuscript Title: Reproducible brain-wide association studies require thousands of individuals

Reviewer Comments & Author Rebuttals

Reviewer Reports on the Initial Version:

Referee #1 (Remarks to the Author):

This paper reports a massive set of analyses across three large neuroimaging datasets, with the aim of understanding the degree to which associations between brain structure/function and other phenotypes are reproducible. The first important finding is that with large samples the observed effect sizes relating brain and behavior are much smaller than those often reported in small-sample studies in the literature. The second important finding is that results from smaller studies (subsampling from the larger dataset) are highly variable and weakly reproducible. The authors further show that the situation is slightly better with multivariate prediction methods, but not qualitatively different.

The results presented here are essential reading for anyone who is interested in assessing associations with any high-dimensional data type, including but not limited to neuroimaging data. From that standpoint I would love to see them published in a high profile journal.

However, I do have a major concern, which is that none of the results presented herein would surprise anyone who has a strong appreciation of statistical theory - at least not anyone who has been paying attention to the discussions over the last decade of the "reproducibility crisis". The only real utility of the neuroimaging data is to calibrate the assessment of the distribution of effect sizes; once that is determined, then one could easily simulate most of the other findings. Many papers over the last decade have presented similar findings, including Button's 2013 analyses of power and effect sizes, Schonbrodt & Perugini's 2013 analyses of correlation stability, and Varoquaux's 2018 analyses of the variability of predictive accuracy as a function of sample size (unfortunately not cited by the authors, though highly relevant). The arguments discussed here are also basically the same ones made in genetics in the context of candidate gene vs. GWAS approaches.

The only substantive criticism that I have regards the power analyses reported in Figure 3. It seems that power was assessed by quantifying the ability to detect those effects that were statistically significant in the full sample at a particular threshold. However, this is not a true measure of power, since this will undoubtedly include true as well as false positives. I am not sure what it should be called, but "power" doesn't seem appropriate unless it's properly qualified in the text.

The lack of availability of the code used to analyze the data for the paper is problematic. I don't expect that there would be any major issues, given how most of the results fit with what would be expected, but it seems to ironically violate basic norms of the kind of reproducible science that the authors are promoting in their paper.

Minor comments:

- p. 4: "median neuroimaging study: N=25" - should provide a source as well as a time period for this number
- p. 9: "Pervasive Statistical Errors" - I initially read this as implying errors in statistical practice, rather than sampling error. It might be good to be clearer about that.
- Figure 3E is potentially confusing since it doesn't have the same X axis scale as the other subpanels.
-

Referee #2 (Remarks to the Author):

In this study, Marek et al., have used three large neuroimaging datasets including ABCD, UK biobank, and HC with a sample size of around ~50000 to investigate the effects of associations between individual differences in brain functions and structures and cognitive and mental health phenotypes. They showed that the BWAS associations were smaller than expected by previous neuroimaging studies, with the univariate correlation median $r = 0.015$. They suggest that such smaller associations will result in replication failures at a typical sample size of $N = 25$. Considering such small effects on BWAS analysis, BWAS require more samples for replication. This manuscript is clearly written and summarizes the results well. The statistical analysis is rigorous, and the findings are solid. However, I have some substantial comments that, if addressed, would enhance the manuscript's usefulness and impact.

1. In the abstract, the authors said three large datasets with $N \sim 50000$ were used in the present study. However, for matching the datasets, they only used a subset of UKB data with a sample size = 877. It would be interesting if the authors can perform their analysis on the whole UKB data to see when the probability of replication, statistical errors, or sampling variability becomes flat (as shown in figure 3).

2. In the introduction, references are missing for some sentences. For example, "Classically univariate, BWAS have recently been facilitated by more powerful, but harder to interpret

multivariate techniques (e.g., support vector regression [SVR], canonical correlation analysis [CCA])..." and "While these open science efforts are improving reproducibility, reliance on relatively small samples (median neuroimaging study: N=25) may have been one of the strongest contributors to BWAS replication failures...". Minor comment, some multivariate approaches (like ICA) are used because they tend to be more interpretable than CCA or PCA, one could even argue the focus on temporally (or subject-wise) coherent networks is actually more interpretable than ROI-based studies. Thus I don't know that I would agree with a blanket statement like the above, though I think I know what the authors mean.

3. ABCD release 2.0 provides both baseline and second-year followup MRI data. It is not clearly stated that which session was used in the present study. Also, each session has multiple scans, and the authors did not provide information about how they used these different scans. It will be interesting that the results can be replicated across scans within each session.

4. In Figures 1 A and B, the authors overlap the plots for Vertex, ROI, and Network, making it hard to visualize their difference and similarity.

5. The authors claimed that more robust BWAS effects were detected for fMRI. This is not fully justified in their findings, especially for the univariate analysis.

6. For the task data analysis, the authors concatenated task data to the resting data for the calculation of RSFC. The RSFC can be calculated for each task condition respectively to further explore whether functional features under different task conditions can provide different effects for the BWAS analysis.

7. The preprocessing of task fMRI was the same as rest fMRI. However, temporal filtering (especially low pass filtering) is not a typical preprocessing step for task data.

8. The authors claim that their focus is mainly on the bivariate associations, and thus they did not consider covariate adjustment. However, covariate adjustment is one of the most important parts in most of the BWAS studies. Age, gender, education, etc. will significantly influence the variability of the features. Also, the sibling, twins and triplets structures of ABCD will also have an impact on the associations. Instead of using correlation analysis, I would recommend using a linear mix effect model that has been widely used in other ABCD studies, which can control for both random effects and fixed effects.

Referee #3 (Remarks to the Author):

A. This paper is to try to quantify the effect of the sample size on the effect sizes and reproducibility of brain-wide association studies. The authors consider the three largest neuroimaging datasets, including HCP, UKB, and ABCD. As expected, the key message is that the sample size should be relative big, requiring thousands of individuals.

B. Although it has several findings, this reviewer has several major concerns about this paper. This paper falls short on several key factors.

C. This reviewer feels that there several important factors that may strongly influence the reproducibility of BWAS. Specifically, I will elaborate on this point below. The first factor is the reproducibility of brain measures (e.g., cortical thickness, TBSS measures, fMRI measures) within individual subjects. It is expected that a highly reproducible measure would lead to better the reproducibility in the downstream BWAS. The reproducibility of brain measures strongly depends on the various “noise components” in the MRI images, such as scanner variation, head motion, and acquisition protocols, among others. Without clearly understanding this key factor, all analyses in this paper are not very informative at all, since one cannot really transfer the knowledge here to the future practices. The second factor is to understand possible “causal” relationships between brain measures, various noise components, and the predictor(s) of interest. Understand such relationships requires both numerical and theoretical analyses. For instance, it would be interesting to statistically derive the related formula for both effect sizes and reproducibility as a function of the sample size at least under some simple models considered in this paper. Different BWASs have different confounding factors and different signal-to-noise ratios, leading to different effect sizes and reproducibility.

D. This paper is lack of appropriate use of statistics and treatment of uncertainties.

E. Conclusions are expected, but they are not quite reliable.

F. Please address point C. A related paper is Dr. Smith's confounding modelling for UBK.

Referee #4 (Remarks to the Author):

The authors conduct brain-wide association analyses of multiple brain-behaviour associations, using large, publicly-available datasets, in order to estimate likely effect sizes for such associations, and the sample sizes necessary to obtain robust results. They also explore the role of different design parameters (e.g., functional vs structural imaging). The conclusion is that robust brain-wide association studies of this kind will require very large sample sizes (by the standard of the field).

Many elements of this study are not in themselves novel – for example, the demonstration that small sample sizes generated variability in effect estimates generated, and the application of a filter for declaring discovery and proceeding to publication (e.g., a significance threshold) leads to effect size inflation in the published literature. However, these elements are worth demonstrating in this context, and together provide a compelling case for using larger samples.

One point that could perhaps be made more strongly is that some of these problems flow from applying a threshold – i.e., relying on statistical significance. A focus on estimation and precision instead may go some way to addressing this, and would place greater emphasis on the magnitude of effects observed. This in turn might encourage more thought as to what would constitute a clinically

or biologically significant effect (as opposed to a merely statistically significant one). With larger sample sizes, this will be particularly important.

Overall, though, this is a very thorough treatment of an important topic, and will provide a rich resource for the field. For example, the inclusion of different types of error (Type 1 and Type 2, but also Type M and Type S), and the demonstration of how these are patterned by sample size is striking. However, I would like to see more discussion of how these issues could be addressed practically. Large sample sizes are available, but bring limitations (especially for functional imaging, where a limited number of tasks may be included).

One possible solution is that we (as researchers, but also with support from funders) move away from an individualistic model of research to a more collaborative one. This is already happening to some extent, and is already an established mode of working in some fields (e.g., genomics). However, there are additional complexities in brain imaging (different technologies, analytical approaches, tasks of interest in the case of functional imaging). These are not intractable problems, but will require cooperation. How can we do this?

As a minor note, the authors mention genome-wide association studies as a methodology associated with poor reproducibility (or at least the sentence could be taken that way). I would suggest that candidate gene studies demonstrated poor reproducibility, but genome-wide association studies have largely generated very robust findings (a combination of large sample size, in-built replication, stringent statistical criteria for claiming discovery, etc.).

I note that the manuscript code will be made available on GitHub on acceptance. I would urge the authors to ensure that details of this (e.g., a URL) are included in the final published version of the manuscript (they may intend to, but this wasn't entirely clear).

I sign my reviews.

Marcus Munafò

Author Rebuttals to Initial Comments:

Referee #1 (Remarks to the Author):

This paper reports a massive set of analyses across three large neuroimaging datasets, with the aim of understanding the degree to which associations between brain structure/function and other phenotypes are reproducible. The first important finding is that with large samples the observed effect sizes relating brain and behavior are much smaller than those often reported in small-sample studies in the literature. The second important finding is that results from smaller studies (subsampling from the larger dataset) are highly variable and weakly reproducible. The authors further show that the situation is slightly better with multivariate prediction methods, but not qualitatively different.

The results presented here are essential reading for anyone who is interested in assessing associations with any high-dimensional data type, including but not limited to neuroimaging data. From that standpoint I would love to see them published in a high profile journal.

Thank you for this positive assessment and classifying the work as 'essential reading'.

However, I do have a major concern, which is that none of the results presented herein would surprise anyone who has a strong appreciation of statistical theory - at least not anyone who has been paying attention to the discussions over the last decade of the "reproducibility crisis". The only real utility of the neuroimaging data is to calibrate the assessment of the distribution of effect sizes; once that is determined, then one could easily simulate most of the other findings. Many papers over the last decade have presented similar findings, including Button's 2013 analyses of power and effect sizes, Schönbrodt & Perugini's 2013 analyses of correlation stability, and Varoquaux's 2018 analyses of the variability of predictive accuracy as a function of sample size (unfortunately not cited by the authors, though highly relevant). The arguments discussed here are also basically the same ones made in genetics in the context of candidate gene vs. GWAS approaches.

Thank you for this thoughtful comment. We agree completely with your assessment. Unfortunately, 'strong appreciation of statistical theory' is less widespread than one would like. Despite important modeling work and discoveries in disciplines outside neuroimaging, reports of inflated BWAS effects derived from very small samples were becoming more prevalent, not less. It seems that there was and is strong hope that the large reported BWAS effect sizes are true or at least only minimally inflated. Thus, a comprehensive demonstration of the true limits of BWAS effect sizes, utilizing much larger data sets that have only recently become available, is essential for bringing about a course correction in BWAS methodology. A striking demonstration backed by massive evidence specific to BWAS seems to be required to actually change current BWAS practices. Button *et al.* and Schönbrodt *et al.* were cited in the original manuscript. We have added the Varoquaux *et al.*, 2018 citation and revised the Introduction and Discussion in line with this comment.

Prior to the release of the ABCD and UK Biobank data, there were no samples large enough to estimate BWAS effect sizes with accuracy, thus limiting prior publications to simulations using different hypothetical effect sizes. The current work is so impactful because it is the first to report empirical BWAS effect sizes from consortium samples that are >2 orders of magnitude larger than typical studies (Figure 1 below). A commentary speculating that many brain-behavioral trait correlations may be inflated (Yarkoni 2009), unfortunately did not prevent all BWAS conducted since (~7% of the annual NIH budget [\$2.8B]; 5,517 active NIH grants: <https://projectreporter.nih.gov/>),

other than a handful, to be massively underpowered. Knowing the true BWAS effect sizes, should now push BWAS research and funding towards greater reproducibility.

Figure 1. Neuroimaging Sample Sizes. The overwhelming majority of BWAS studies contain < 100 subjects. The histogram shows sample sizes for neuroimaging studies uploaded to the OpenNeuro database (blue), BWAS sample sizes of depression (red) from a 2017 meta-analysis by (Müller et al. 2017), and BWAS sample sizes of IQ (green) from. The revised manuscript utilizes the three largest neuroimaging samples to date: HCP (N~1,200), ABCD (N~12,000), and UK Biobank (N~37,000).

Unfortunately, the notion that sample sizes of ~25 are a thing of the past, as they should be, is not supported by data. We conducted a sample size analysis of brain MRI data sets currently available for download at openneuro.org, which revealed a median sample size of N=21 (Figure 1). A recent meta-analysis of inter-individual differences neuroimaging studies in Major Depression (Figure 1, DepMeta) found a median sample size of N=31 (Müller et al. 2017), another meta-analysis of IQ studies (Figure 1, IQMeta) had N=44 as the median sample size.

The only substantive criticism that I have regards the power analyses reported in Figure 3. It seems that power was assessed by quantifying the ability to detect those effects that were statistically significant in the full sample at a particular threshold. However, this is not a true measure of power, since this will undoubtedly include true as well as

false positives. I am not sure what it should be called, but "power" doesn't seem appropriate unless it's properly qualified in the text.

Thank you for this comment. As you correctly noted, there is a subtle, but important distinction between classical statistical power (analytically derived through a program such as G*Power) and the bootstrapped approach underlying Figure 3. Whereas a classical power analysis utilizes an analytic/formulaic calculation to derive statistical power (1-false negative rate) for simple univariate models, our approach which was influenced by prior simulation work (Schönbrodt and Perugini, 2013) utilized a bootstrapped procedure to compare “sub-samples” (simulated smaller studies) to a population estimate. This approach had the advantages of 1) being non-parametric in the estimation of the sampling distribution, and 2) that it could be repeated identically in multivariate analyses, for which analytic/formulaic classical power analyses do not exist. In the revised manuscript we have clarified the wording and made a distinction between this method of assessing “stabilization of effect size” as in Schönbrodt and Perugini, 2013 via bootstrapped subsampling and classical “analytic statistical power estimates” through G*Power. The revised text is as follows:

“To quantify how the pairing of smaller-than-expected effect sizes and sampling variability (i.e., random variation of an association across population subsamples) impacts BWAS reproducibility, we characterized the relationship between statistical errors and sample size across significance thresholds ($p < 0.05$ - $p < 10^{-7}$) using non-parametric bootstrapping²⁰ (Fig. 3) and verified the results with analytic statistical power estimations (Fig. S15; Fig. S16 for UKB)³⁴.“

In addition, the revised manuscript more clearly highlights that supplemental, classical statistical power calculations using standard analytic solutions (i.e., G*Power, Fig. S15) provide the same results as those shown in Figure 3. Below we include Fig. S15 in the revised manuscript, which depicts a G*Power power estimate across sample sizes for an $r=0.06$ (top 1% of BWAS effect sizes) across sample sizes for $p < 0.05$ and $p < 10^{-7}$ (Bonferroni correction).

Fig. S15. Relationship between Statistical Power and Sample Size. Analytic requisite sample sizes to detect 99th percentile (largest 1%; $r=0.06$) univariate brain-wide associations at **(A)** $p < 10^{-7}$ ($p < 0.05$ Bonferroni corrected) and **(B)** $p < 0.05$ (uncorrected).

The lack of availability of the code used to analyze the data for the paper is problematic. I don't expect that there would be any major issues, given how most of the results fit with what would be expected, but it seems to ironically violate basic norms of the kind of reproducible science that the authors are promoting in their paper.

We apologize for not having the code made available at the time of the initial submission. We have now uploaded the code to Gitlab now that the paper is in revision. The analysis code is available here: <https://gitlab.com/DosenbachGreene/bwas>

Minor comments:

- p. 4: "median neuroimaging study: N=25" - should provide a source as well as a time period for this number

The source (OpenNeuro) and time period (all records up until September 2021) have been added to the revised manuscript. Thank you for noticing this omission.

- p. 9: "Pervasive Statistical Errors" - I initially read this as implying errors in statistical practice, rather than sampling error. It might be good to be clearer about that.

Thank you for pointing this out. This point has been clarified and the corresponding subheading has been changed from '*Pervasive Statistical Errors Undermine BWAS Reproducibility*' to '*High Statistical Error Rates Undermine BWAS Reproducibility*'.

- Figure 3E is potentially confusing since it doesn't have the same X axis scale as the other subpanels.

Great point. We have amended Figure 3 to have consistent X-axes and Y-axes to promote harmony in comparing across figure panels. We note in the Panel E figure caption that data ends at N~2,000 (½ full sample), since non-overlapping discovery and replication samples are used to estimate replicability (i.e., half of full sample).

Referee #2 (Remarks to the Author):

In this study, Marek et al., have used three large neuroimaging datasets including ABCD, UK biobank, and HC with a sample size of around ~50000 to investigate the effects of associations between individual differences in brain functions and structures and cognitive and mental health phenotypes. They showed that the BWAS associations were smaller than expected by previous neuroimaging studies, with the univariate correlation median $r = 0.015$. They suggest that such smaller associations will result in replication failures at a typical sample size of $N = 25$. Considering such small effects on BWAS analysis, BWAS require more samples for replication. This manuscript is clearly written and summarizes the results well. The statistical analysis is rigorous, and the findings are solid. However, I have some substantial comments that, if addressed, would enhance the manuscript's usefulness and impact.

Thank you for highlighting the strength and clarity of the analyses, findings and writing.

1. In the abstract, the authors said three large datasets with $N \sim 50000$ were used in the present study. However, for matching the datasets, they only used a subset of UKB data

with a sample size = 877. It would be interesting if the authors can perform their analysis on the whole UKB data to see when the probability of replication, statistical errors, or sampling variability becomes flat (as shown in figure 3).

We agree completely with this suggestion. The revised manuscript now utilizes all available UKB data (N= 32,572; see Supplemental Fig. S16 - included below for convenience) Results from the full UKB sample were consistent with those from the UKB subsample and the full ABCD and HCP samples.

Fig. S16. Statistical Errors and Reproducibility of Univariate Brain-Wide Associations in the UK Biobank (UKB). (A) False negative rates for correlations between fluid intelligence and edgewise RSFC (y-axis), as a function of sample size (x-axis) and p-value (color scale). The most stringent p-value (10^{-7}) is equivalent to a Bonferroni correction across all RSFC pairs. (B) Magnitude error likelihoods for three levels of effect size inflation (50%, 100%, 200%) shown for correlations between edgewise RSFC (y-axis), as a function of sample size (x-axis) and statistical threshold ($p < 0.05$, $p < 10^{-7}$). Color gradient represents the magnitude of the effect size inflation. Solid lines represent inflation for effects that survive statistical thresholding at $p < 0.05$. Dotted lines represent inflation for effects that survive statistical thresholding at $p < 10^{-7}$. (C) Sign error likelihoods representing the probability that a subsample gives the opposite sign compared to the full sample, for correlations between fluid intelligence and edgewise RSFC (y-axis), as a function of sample size (x-axis) and p-value (color scale). Dotted line represents sign error rates regardless of statistical thresholding. (D) Statistical power for correlations between fluid intelligence edgewise RSFC (y-axis), as a function of sample size (x-axis) and p-value (color scale). (E) Probability (%) of replicating a univariate brain-phenotype association in the replication sample that was deemed significant in the discovery sample (note: data ends at $N \sim 15,000$ since the replication sample is approximately half of the full sample). Multiple significance thresholds (p-values) were surveyed (color

scale). (F) False positive rates for correlations between fluid intelligence and edgewise RSFC (y-axis), as a function of sample size (x-axis) and p-value (color scale).

2. In the introduction, references are missing for some sentences. For example, “Classically univariate, BWAS have recently been facilitated by more powerful, but harder to interpret multivariate techniques (e.g., support vector regression [SVR], canonical correlation analysis [CCA])...” and “While these open science efforts are improving reproducibility, reliance on relatively small samples (median neuroimaging study: N=25) may have been one of the strongest contributors to BWAS replication failures...”. Minor comment, some multivariate approaches (like ICA) are used because they tend to be more interpretable than CCA or PCA, one could even argue the focus on temporally (or subject-wise) coherent networks is actually more interpretable than ROI-based studies. Thus I don't know that I would agree with a blanket statement like the above, though I think I know what the authors mean.

Thank you for these helpful comments. The missing references have been added to the revised version of the manuscript. We have also clarified the sentence about the interpretability of multivariate analyses referenced here.

3. ABCD release 2.0 provides both baseline and second-year followup MRI data. It is not clearly stated that which session was used in the present study. Also, each session has multiple scans, and the authors did not provide information about how they used these different scans. It will be interesting that the results can be replicated across scans within each session.

The revised Methods section now includes additional clarification of these points. The manuscript is based on session 1 (baseline) ABCD data. In the baseline dataset (as well as subsequent years), the ABCD Study attempted to collect 4 x 5 minute resting state BOLD scans per participant. For each participant, we used all available data that passed quality control. The manuscript specifically evaluates cross-sectional individual difference studies. FastTrack (whole brain, non-tabulated) ABCD year 2 imaging data have yet to be processed/released and will be the focus of subsequent work. The revised Discussion also makes more explicit mention of the potential benefits of longitudinal designs, which do not fall under the definition of BWAS.

4. In Figures 1 A and B, the authors overlap the plots for Vertex, ROI, and Network, making it hard to visualize their difference and similarity.

We compressed this information for the main manuscript, but have also generated a version of this information that shows the histograms for vertices, ROIs and networks separately (Fig. S2, see below for convenience).

Fig. S2. Histograms of all (A) cortical thickness associations with cognitive ability (green) and psychopathology (purple) and (B) RSFC associations with cognitive ability (green) and psychopathology (purple). Phenotypic correlations with brain measures were generated across multiple levels of scale (cortical thickness: vertices, ROIs, networks; RSFC: ROI-ROI pairs (edges), principal components, networks), which are represented in varying shades of green and purple to match Fig. 1A,B. Each paneled histogram contains the same x-axis as the upper leftmost panel in (A).

5. The authors claimed that more robust BWAS effects were detected for fMRI. This is not fully justified in their findings, especially for the univariate analysis.

We apologize for not being more direct about this. For univariate BWAS, the top 1st percentile of associations between cortical thickness and cognitive ability and psychopathology was $r > 0.06$, whereas the top 1st percentile of associations between RSFC and cognitive ability and psychopathology was $r > 0.08$. The original manuscript visualized the relatively greater BWAS robustness/reproducibility of functional compared to structural MRI most clearly for multivariate analyses (i.e., Figure 4E). While Figure 1 A-D shows the univariate effect size distributions side-by-side, we never directly contrasted the reproducibility (similarity of effect size between discovery and replication datasets) in a single plot. To more clearly visualize that the same pattern of relative BWAS reproducibility holds for univariate and multivariate analyses, we have added additional plots to the revised supplement that directly contrast the strongest (top 1%, $\sim r > 0.06$) effect sizes in the full discovery vs. replication datasets for fMRI vs. cortical thickness and cognition vs. psychopathology (Fig. S20).

Fig. S21. Inflation of Univariate Effect Sizes (Top 1% Largest) by Imaging Modality and Behavioral Phenotype. Better out-of-sample replication is indexed by a smaller difference between the discovery and replication sample (right side of histogram). Negative values indicate that an association was inflated in the Discovery dataset. Values $<-100\%$ indicate an association sign difference between Discovery and Replication data sets (sign error). The leftward shift of the distribution in cortical thickness relative to RSFC and psychopathology relative to cognitive ability indicates worse univariate BWAS reproducibility.

6. For the task data analysis, the authors concatenated task data to the resting data for the calculation of RSFC. The RSFC can be calculated for each task condition respectively to further explore whether functional features under different task conditions can provide different effects for the BWAS analysis.

We have previously investigated the effects of different tasks on RSFC (Gratton *et al.*, Neuron, 2018). Our prior work, as well as similar studies by others (Chen *et al.*, 2020, BioRxiv), revealed that task effects on resting state functional connectivity are extremely small ($<2\%$ variance explained).

In this manuscript, as the reviewer correctly noted, we did concatenate task data to rest data in the ABCD dataset, which did not result in larger effect sizes. However, we also determined BWAS effect sizes across 86 task contrasts and 39 behavioral phenotypes (Tables S4 and S5) from the Human Connectome Project dataset. The BWAS effect size distributions were equivalent to RSFC across task conditions. Specifically, the ABCD and HCP included different tasks, but the BWAS effect size histograms when including task fMRI data, were the same. That said, further investigations of task differences in BWAS may well be the subject of follow-on publications.

7. The preprocessing of task fMRI was the same as rest fMRI. However, temporal filtering (especially low pass filtering) is not a typical preprocessing step for task data.

That is correct. While temporal filtering and other preprocessing steps are not typical for task fMRI data, there is ample precedence for doing so. Prior work has suggested that applying resting state functional connectivity preprocessing to task fMRI data improves the replicability of task fMRI results (Siegel *et al.*, 2014, NeuroImage). For this manuscript we chose to pre-process the resting state and task BOLD data identically to exclude pre-processing choices from the list of potential contributors to poor reproducibility (Gratton *et al.*, 2018, Neuron). We processed data identically to have an apples-to-apples comparison between resting- and task-states. Otherwise, we would be unable to disentangle differences in biology (rest vs. task) from differences arising from methodological variability (choice in processing, e.g., Botvinik-Nezer *et al.*, 2020, *Nature*).

8. The authors claim that their focus is mainly on the bivariate associations, and thus they did not consider covariate adjustment. However, covariate adjustment is one of the most important parts in most of the BWAS studies. Age, gender, education, etc. will significantly influence the variability of the features. Also, the sibling, twins and triplets structures of ABCD will also have an impact on the associations. Instead of using correlation analysis, I would recommend using a linear mix effect model that has been widely used in other ABCD studies, which can control for both random effects and fixed effects.

We fully concur. As suggested, we carried out linear mixed effects modeling using the set of covariates created by the ABCD Study analysis core and used in the official NIMH ABCD Data Exploration and Analysis Portal (<https://deap.nimhda.org>; Dick *et al.*, 2021, NeuroImage) developed by Wes Thompson (manuscript co-author). The strongest (top 1%) BWAS effects are on average further diminished by 25% when accounting for socio-demographic covariates (Fig. S4). The revised manuscript now makes mention of this and the results of the covariate analyses are included in the revised supplement (Fig. S4 - see below for convenience).

Fig. S4. Impact of Sociodemographic Covariates on BWAS Effect Sizes. The influence of sociodemographic covariates (race, gender, parental marital status, parental income, hispanic versus non hispanic ethnicity, family, data collection site) on BWAS effect sizes was examined in the ABCD dataset through the model comparison strategy develop by the ABCD Data Analysis and Informatics Core and used in the Data Exploration and Analysis Portal (deap.nimhda.org). The percentages of variance explained by fixed effects in multilevel models (pseudo R^2) were calculated with the MuMIn package in R and square root transformed to approximate an absolute-value BWAS correlation coefficient (r). The estimated BWAS effect sizes (r) prior to covariate adjustment are plotted on the x-axis and those after sociodemographic covariate adjustment on the y-axis. Values below the identity line indicate a reduction in effect size after covariate adjustment, values above the identity line indicate an increase in effect size after covariate adjustment. BWAS models with and without covariate adjustment always included cognitive ability or psychopathology as the outcome variable and nested random effects of family and data collection sites, in order to maximize comparability for subsequent fixed effect model comparisons. BWAS effect sizes without covariate adjustment were taken from models that only included these random effects, the brain feature of interest (cortical thickness [vertex]/RSFC [edge]) as a single fixed effect, and the outcome (cognitive ability/psychopathology). BWAS effect sizes without covariate adjustment estimated the unique, covariate adjusted effect linking the brain feature of interest (cortical thickness [vertex]/RSFC

[edge]) to the outcome (cognitive ability/psychopathology) by comparing a model with sociodemographic fixed effects but no brain feature fixed effect, to one with both the sociodemographic fixed effects and the brain feature. Thus, this difference in pseudo R^2 (subsequently transformed to r) represented how much additional fixed effect variance the brain feature explained beyond the sociodemographic covariates.

Referee #3 (Remarks to the Author):

A. This paper is to try to quantify the effect of the sample size on the effect sizes and reproducibility of brain-wide association studies. The authors consider the three largest neuroimaging datasets, including HCP, UKB, and ABCD. As expected, the key message is that the sample size should be relative big, requiring thousands of individuals.

Thank you for this excellent synopsis of the manuscript.

B. Although it has several findings, this reviewer has several major concerns about this paper. This paper falls short on several key factors.

C. This reviewer feels that there several important factors that may strongly influence the reproducibility of BWAS. Specifically, I will elaborate on this point below. The first factor is the reproducibility of brain measures (e.g., cortical thickness, TBSS measures, fMRI measures) within individual subjects. It is expected that a highly reproducible measure would lead to better the reproducibility in the downstream BWAS. The reproducibility of brain measures strongly depends on the various “noise components” in the MRI images, such as scanner variation, head motion, and acquisition protocols, among others. Without clearly understanding this key factor, all analyses in this paper are not very informative at all, since one cannot really transfer the knowledge here to the future practices. The second factor is to understand possible “causal” relationships between brain measures, various noise components, and the predictor(s) of interest.

Understand such relationships

requires both numerical and theoretical analyses. For instance, it would be interesting to statistically derive the related formula for both effect sizes and reproducibility as a function of the sample size at least under some simple models considered in this paper. Different BWASs have different confounding factors and different signal-to-noise ratios, leading to different effect sizes and reproducibility.

D. This paper is lack of appropriate use of statistics and treatment of uncertainties.

Measurement Reliability

Thank you for highlighting the importance of measurement reliability when considering reproducibility. The original manuscript contained this brief paragraph discussing the issue:

“Low measurement reliability can attenuate the observed correlation between two variables [38]. Within-subject measurement reliabilities for the exemplar behavioral phenotypes (r : NIHTB = 0.9039; CBCL = 0.9440) and imaging measures (r : cortical thickness >0.9641 ; RSFC: ABCD = 0.48; HCP = 0.79; UKB=0.39; Fig. S10) are moderate to high. Further improvements in functional MRI measurement reliability could theoretically increase BWAS effect sizes slightly (Fig. S11; see Supplemental Discussion). ”

Which pointed to a lengthier discussion in the Supplement that contained the formula for the relationship between measurement error and correlation strength, as well as a series of simulations that showed how much BWAS effects of different sizes could be expected to increase with measurement error reductions. These analyses demonstrated that the imaging metrics with the strongest baseline correlation with behavior (i.e., fMRI) and the relatively highest measurement errors (i.e, fMRI) could theoretically become more strongly correlated with behavior upon further reduction of measurement errors. Such strengthening of the fMRI - behavior correlations is only theoretical, because the effect size is always bounded by the true underlying biological relationship. For imaging measures more weakly correlated with behavior (i.e., cortical thickness) and very low current measurement error (i.e., cortical thickness), improved measurement will not affect the correlations. Some measures (i.e. cortical thickness) are simply not that strongly related to behavioral measures.

To further highlight this critically important point, we added a sentence to the main manuscript pointing out that while fMRI metrics still contain some measurement error, the structural MRI measures (i.e. cortical thickness) have only minimal measurement error, but correlate less strongly with behavioral measures.

Confounding Factors

We examined the influence of sociodemographic covariates standardly used in ABCD analyses (race, gender, parental marital status, parental income, hispanic versus non hispanic ethnicity, family, data collection site) on BWAS effect sizes noting that the in general decrease effect sizes, particularly for the largest BWAS effects (see Fig. S4, included below for convenience). Sociodemographic covariate adjustment resulted in decreased effect sizes, especially for the strongest associations (top1% Δr =-0.014; Fig. S4).

Fig. S4. Impact of Sociodemographic Covariates on BWAS Effect Sizes. The influence of sociodemographic covariates (race, gender, parental marital status, parental income, hispanic versus non hispanic ethnicity, family, data collection site) on BWAS effect sizes was examined in the ABCD dataset through the model comparison strategy develop by the ABCD Data Analysis and Informatics Core and used in the Data Exploration and Analysis Portal (deap.nimhda.org). The percentages of variance explained by fixed effects in multilevel models (pseudo R^2) were calculated with the MuMIn package in R and square root transformed to approximate an absolute-value BWAS correlation coefficient (r). The estimated BWAS effect sizes (r) prior to covariate adjustment are plotted on the x-axis and those after sociodemographic covariate adjustment on the y-axis. Values below the identity line indicate a reduction in effect size after covariate adjustment, values above the identity line indicate an increase in effect size after covariate adjustment. BWAS models with and without covariate adjustment always included cognitive ability or psychopathology as the outcome variable and nested random effects of family and data collection sites, in order to maximize comparability for subsequent fixed effect model comparisons. BWAS effect sizes without covariate adjustment were taken from models that only included these random effects, the brain feature of interest (cortical thickness [vertex]/RSFC [edge]) as a single fixed effect, and the outcome (cognitive ability/psychopathology). BWAS effect sizes without covariate adjustment estimated the unique, covariate adjusted effect linking the brain feature of interest (cortical thickness [vertex]/RSFC

[edge]) to the outcome (cognitive ability/psychopathology) by comparing a model with sociodemographic fixed effects but no brain feature fixed effect, to one with both the sociodemographic fixed effects and the brain feature. Thus, this difference in pseudo R^2 (subsequently transformed to r) represented how much additional fixed effect variance the brain feature explained beyond the sociodemographic covariates.

E. Conclusions are expected, but they are not quite reliable.

While the results were expected by some, mostly outside of neuroimaging, the overwhelming majority of neuroimaging researchers did not expect them. For this project we sought advice from colleagues in genomics conducting GWAS. This cross-disciplinary collaboration was immensely helpful. Early on, our colleagues in genomics strengthened our resolve to persist in this work, because many of them were not surprised by our findings, given their own research. In sharp contrast other neuroimaging researchers were initially extremely resistant to accepting the somewhat painful results presented in this manuscript. To overcome the strong initial resistance from GWAS researchers and to prove the reliability of the findings, we based the manuscript on the three largest datasets (UKB, ABCD, HCP) currently available, which all provided the same results.

F. Please address point C. A related paper is Dr. Smith's confounding modelling for UKB.

The revised manuscript more clearly addresses point C and references the 2018 Neuron article by Smith and Nichols (manuscript co-author), mentioned here.

Referee #4 (Remarks to the Author):

The authors conduct brain-wide association analyses of multiple brain-behaviour associations, using large, publicly-available datasets, in order to estimate likely effect sizes for such associations, and the sample sizes necessary to obtain robust results. They also explore the role of different design parameters (e.g., functional vs structural imaging). The conclusion is that robust brain-wide association studies of this kind will require very large sample sizes (by the standard of the field).

Many elements of this study are not in themselves novel – for example, the demonstration that small sample sizes generated variability in effect estimates generated, and the application of a filter for declaring discovery and proceeding to

publication (e.g., a significance threshold) leads to effect size inflation in the published literature. However, these elements are worth demonstrating in this context, and together provide a compelling case for using larger samples.

Thank you for this most incisive summary of the manuscript.

One point that could perhaps be made more strongly is that some of these problems flow from applying a threshold – i.e., relying on statistical significance. A focus on estimation and precision instead may go some way to addressing this, and would place greater emphasis on the magnitude of effects observed. This in turn might encourage more thought as to what would constitute a clinically or biologically significant effect (as opposed to a merely statistically significant one). With larger sample sizes, this will be particularly important.

We fully concur with this assessment. The revised Discussion now explores these very important points in greater depth. Specifically:

“As with GWAS⁵⁰, funding agencies should boost the aggregation of BWAS appropriate datasets through mandatory sharing policies. Even for large datasets collected and processed identically, in-sample associations are stronger than out-of-sample replications (Fig. 4E, Fig S21); therefore, reporting both in- and out-of-sample effect sizes should be a publication and funding requirement. BWAS may also benefit from focusing data collection on the most reliable and robust brain-phenotype associations (e.g., functional vs. structural; direct behavioral vs. questionnaire).”

Overall, though, this is a very thorough treatment of an important topic, and will provide a rich resource for the field. For example, the inclusion of different types of error (Type 1 and Type 2, but also Type M and Type S), and the demonstration of how these are patterned by sample size is striking. However, I would like to see more discussion of how these issues could be addressed practically. Large sample sizes are available, but bring limitations (especially for functional imaging, where a limited number of tasks may be included).

Thank you for this comment. The revised manuscript now concludes with a forward-looking section explicitly discussing ways to potentially address the issues raised. We have added the following section to the end of the manuscript:

“Reproducibly Linking Brain and Behavior

All brain-behavior studies will benefit from technological advances that generate higher quality brain and behavioral data with greater efficiency, such as real-time quality control³⁸, multi-band multi-echo sequences⁴⁷ and thermal denoising for fMRI⁴⁸, as well as deep behavioral phenotyping with ecological momentary assessment (EMA) and passive sensing⁴⁹.

As with GWAS⁵⁰, funding agencies should boost the aggregation of BWAS appropriate datasets through mandatory sharing policies. Even for large datasets collected and processed identically, in-sample associations are stronger than out-of-sample replications (Fig. 4E, Fig S21); therefore, reporting both in- and out-of-sample effect sizes should be a publication and funding requirement. BWAS may also benefit from focusing data collection on the most reliable and robust brain-phenotype associations (e.g., functional vs. structural; direct behavioral vs. questionnaire).

The brain, in contrast to the genome, is expected to change over time and can be manipulated ethically. For greater effect sizes and statistical power, neuroscience should focus on within-participant over cross-sectional study designs, and on interventional (therapy, medications, brain stimulation, surgery) over observational ones. Rather than associating pre-defined psychological constructs and brain features⁵¹, data-driven, combined brain-behavior phenotypes will further advance our understanding of cognition and mental health. Hence, our prospects for linking neuroimaging markers to complex human behaviors are better than ever.”

One possible solution is that we (as researchers, but also with support from funders) move away from an individualistic model of research to a more collaborative one. This is already happening to some extent, and is already an established mode of working in some fields (e.g., genomics). However, there are additional complexities in brain imaging (different technologies, analytical approaches, tasks of interest in the case of functional imaging). These are not intractable problems, but will require cooperation. How can we do this?

These questions are absolutely vital to the future of BWAS research. We are now more explicitly suggesting potential solutions with the intent to stirring up a fruitful debate amongst BWAS researchers. We now note that:

“As with GWAS⁵⁰, funding agencies should boost the aggregation of BWAS appropriate datasets through mandatory sharing policies. Even for large datasets collected and processed identically, in-sample associations are stronger than out-of-sample replications (Fig. 4E, Fig S21); therefore, reporting both in- and out-of-sample effect sizes should be a publication and funding requirement. BWAS may also benefit from focusing data collection on the most reliable and robust brain-phenotype associations (e.g., functional vs. structural; direct behavioral vs. questionnaire).”

As a minor note, the authors mention genome-wide association studies as a methodology associated with poor reproducibility (or at least the sentence could be taken that way). I would suggest that candidate gene studies demonstrated poor reproducibility, but genome-wide association studies have largely generated very robust findings (a combination of large sample size, in-built replication, stringent statistical criteria for claiming discovery, etc.).

Thank you for noticing. This error has been corrected.

I note that the manuscript code will be made available on GitHub on acceptance. I would urge the authors to ensure that details of this (e.g., a URL) are included in the final published version of the manuscript (they may intend to, but this wasn't entirely clear).

The code has been further annotated and is shared here:

<https://gitlab.com/DosenbachGreene/bwas>

For code relevant to processing of ABCD and UKB data see:

<https://github.com/DCAN-Labs/abcd-hcp-pipeline>

I sign my reviews.

Marcus Munafò

Reviewer Reports on the First Revision:

Referee #1 (Remarks to the Author):

The authors have responded impressively to my concerns (as well as, by my reading, to the concerns of the other reviewers). I agree with them regarding the high prevalence of badly underpowered studies, and I think that publication of this paper in Nature will help to turn the tide towards more reproducible work in this field.

EDITORIAL NOTE: Reviewer 2 was unable to re-review this version of the manuscript, so Reviewer 1 graciously agreed to look over the responses to R2's comments and indicated that they had been adequately addressed.

Referee #3 (Remarks to the Author):

This reviewer appreciates the amount of efforts that all authors devoted to this revision. As stated in their response letter, most findings presented in this paper are well known in the field of GWASs, even though they have not been widely acceptable in the neuroimaging community due to the fact that most existing studies cannot afford the collection of thousands of subjects. After reading the revision, this reviewer still has several major comments.

First, HCP contains many twins. Throughout the paper, the authors do not account for the twin structure in HCP throughout the paper, since such twin structure can confound all statistical results presented in the paper. Specifically, the standard bootstrap method used here is not quite valid here.

Second, in lines 27-28 on page 6, “Smaller brain-wide association studies have reported larger univariate correlation ($r > 0.2$) than the largest effects we measured in much larger samples. How could it be?” Although the authors correctly stated that the sample size is one possible reason, but it is not the only reason. Smaller sample size only introduces larger variability, whereas the strong relationship does play an important role here.

Third, in the second paragraph of page 7, the authors reported the measurement reliability for imaging measures, such as moderate reliability metrics for RSFC. In the recent report by the same (smaller) group of authors (“Reliability and Stability Challenges in ABCD Task fMRI Data”), the authors reported much smaller ICC values. The measurement reliability cannot be reduced by using larger sample size. Could the authors elaborate more on this point?

Fourth, most replication(r)-discovery(r) values are negative throughout the paper. Could the authors elaborate more the underlying statistical theory for this phenomenon?

Fifth, in lines 2-3 on page 12, the authors stated that minimum sample size requirements depend on the study aims. This statement cannot be true, since many factors can contribute the sample size requirements.

Referee #4 (Remarks to the Author):

I made a number of relatively minor recommendations in the previous round, but these were largely around the need for a forward-looking discussion of how some of the implications of this work could be addressed practically.

The authors have done this in a thorough and thoughtful way. Overall, I am confident that this work will make an important contribution, and I hope it will be read and taken seriously by researchers, funders and journals.

Author Rebuttals to First Revision:

Referee #3 (Remarks to the Author):

This reviewer appreciates the amount of efforts that all authors devoted to this revision. As stated in their response letter, most findings presented in this paper are well known in the field of GWASs, even though they have not been widely acceptable in the neuroimaging community due to the fact that most existing studies cannot afford the collection of thousands of subjects. After reading the revision, this reviewer still has several major comments.

First, HCP contains many twins. Throughout the paper, the authors do not account for the twin structure in HCP throughout the paper, since such twin structure can confound all statistical results presented in the paper. Specifically, the standard bootstrap method used here is not quite valid here.

Similar to HCP, ABCD also contains twins and siblings. Early in our analyses of the effect sizes and sampling variability of BWAS we considered the effects of twins. We ran peripheral analyses and determined that the effect size estimates and sampling variability (quantified using a bootstrap

approach of resampling with replacement) were the same regardless of whether or not siblings/twins were included. Specifically, the effect size distribution (top 1% of effect sizes were $\sim r=0.08$ regardless) and sampling variability across all associations were nearly identical (see Fig. R1; average difference = 0.0005). Given that the effect sizes and sampling variability were unchanged, we elected to include twins and siblings in the final sample to boost our sample size as much as possible. We have now included this analysis in the Methods and we provide the figure below as Extended Data Fig. 10.

Fig. R1. Sampling variability is nearly identical when considering singletons vs. all participants. Data were from the ABCD Study sample. Sampling variability (y-axis) as a function of sample size (x-axis) ranging from $n = 25$ -2,000 for all participants (black) and singletons only (twins and siblings excluded; green). Sampling variability was quantified as the difference between the upper and lower 95% confidence interval across 1,000 bootstraps (resampled with replacement) across all 77,421 resting-state functional connectivity (RSFC) edges.

In all replication analyses, the established Discovery and Replication samples (e.g., Fig. 4), developed by the ABCD consortium and used in prior work, were matched such that twin pairs were always kept together (i.e., each sibling within a twin pair would be in the Replication sample or Discovery sample and cannot be split). Thus, twin pairs could not artificially inflate the effect sizes. In addition, the existing manuscript includes supplemental analyses with ABCD data that included nested random effects of family and data collection site along with salient fixed effect demographic factors (race, gender, parental marital status, parental income, Hispanic; see Extended Data Fig. 2). These supplemental analyses demonstrated that modeling these factors further reduced BWAS effect sizes, thus strengthening the primary conclusions of the manuscript.

Second, in lines 27-28 on page 6, “Smaller brain-wide association studies have reported larger univariate correlation ($r>0.2$) than the largest effects we measured in much larger samples. How could it be?” Although the authors correctly stated that the sample size is one possible reason, but it is not the only reason. Smaller sample size only introduces larger variability, whereas the strong relationship does play an important role here.

We strongly agree that sample size may not be the only reason, and in the Introduction state: “Factors that have contributed to poor reproducibility of population-based research in psychology¹, genomics², and medicine³, such as methodological variability⁴, data mining for significant results⁵, overfitting⁶, confirmation and publication biases⁷, and inadequate statistical power⁸ likely also affect BWAS.”

In the Summary paragraph we underline the importance of BWAS effect sizes: “Smaller than expected brain-phenotype associations and variability across population subsamples can explain widespread BWAS replication failures.” In the introduction (p.4, line 29-30) we again highlight the importance of effect sizes: “Thus if true brain-wide associations were smaller than previously assumed ($r=0.2-0.8$), larger samples would be required to accurately measure them”.

Third, in the second paragraph of page 7, the authors reported the measurement reliability for imaging measures, such as moderate reliability metrics for RSFC. In the recent report by the same (smaller) group of authors (“Reliability and Stability Challenges in ABCD Task fMRI Data”), the authors reported much smaller ICC values. The measurement reliability cannot be reduced by using larger sample size. Could the authors elaborate more on this point?

In the manuscript referenced by the reviewer⁹, the reliability in question is for task fMRI activations, whereas the current manuscript reports measurement reliability for structural MRI and resting-state functional connectivity (RSFC). Due to use of a subtraction method, where difference scores propagate error from both conditions¹⁰, the measurement reliability of task fMRI activation contrasts is lower than that of RSFC and structural MRI.

Fourth, most replication(r)-discovery(r) values are negative throughout the paper. Could the authors elaborate more the underlying statistical theory for this phenomenon?

Systematically reduced replication effect sizes have also been documented in psychology¹¹ and cancer biology¹², where effect sizes deflated by 80% and 85%, respectively, in replication samples, relative to the published statistically significant discovery effect sizes. This reflects a selection bias for inflated effect sizes when (e.g., significant p-value; strongest associations) and a subsequent regression to the mean (true-population effect size) on replication^{12,13}. This phenomenon is discussed under the heading “The Underpowered BWAS Paradox”, where we state “when attempting to replicate inflated BWAS associations, regression to the mean (actual effect size) makes non-significance (i.e., replication failure) the most likely outcome” (p.11 34-36).

Fifth, in lines 2-3 on page 12, the authors stated that minimum sample size requirements depend on the study aims. This statement cannot be true, since many factors can contribute the sample size requirements.

Many factors contribute to BWAS sample size requirements and the manuscript reports results to this end (e.g.: multivariate vs. univariate analyses; measurement reliability). The sentence in question and the surrounding paragraph discuss the relative sample size differences between BWAS and other types of neuroimaging studies. For clarification we have replaced 'aims' with 'designs'.

References

1. Nosek, B. A., Cohoon, J., Kidwell, M. & Spies, J. R. Estimating the Reproducibility of Psychological Science. doi:10.31219/osf.io/447b3.
2. Visscher, P. M. *et al.* 10 Years of GWAS Discovery: Biology, Function, and Translation. *Am. J. Hum. Genet.* **101**, 5–22 (2017).
3. Begley, C. G. & Ellis, L. M. Drug development: Raise standards for preclinical cancer research. *Nature* **483**, 531–533 (2012).
4. Botvinik-Nezer, R. *et al.* Variability in the analysis of a single neuroimaging dataset by many teams. *Nature* **582**, 84–88 (2020).
5. Nuzzo, R. Scientific method: Statistical errors. *Nature* vol. 506 150–152 (2014).
6. Hawkins, D. M. The problem of overfitting. *J. Chem. Inf. Comput. Sci.* **44**, 1–12 (2004).
7. Bishop, D. How scientists can stop fooling themselves over statistics. *Nature* vol. 584 9–9 (2020).
8. Button, K. S. *et al.* Power failure: why small sample size undermines the reliability of neuroscience. *Nat. Rev. Neurosci.* **14**, 365–376 (2013).
9. Kennedy, J. T. *et al.* Reliability and Stability Challenges in ABCD Task fMRI Data. doi:10.1101/2021.10.08.463750.
10. Rogosa, D. R. & Willett, J. B. DEMONSTRATING THE RELIABILITY THE DIFFERENCE SCORE IN THE MEASUREMENT OF CHANGE. *Journal of Educational Measurement* vol. 20 335–343 (1983).
11. Patil, P., Peng, R. D. & Leek, J. T. What Should Researchers Expect When They Replicate Studies? A Statistical View of Replicability in Psychological Science. *Perspect. Psychol. Sci.* **11**, 539–544 (2016).
12. Errington, T. M., Denis, A., Perfito, N., Iorns, E. & Nosek, B. A. Challenges for assessing replicability in preclinical cancer biology. *Elife* **10**, (2021).
13. Held, L., Pawel, S. & Schwab, S. Replication power and regression to the mean. *Significance* vol. 17 10–11 (2020).

EDITORIAL NOTE: We sent these responses back to Reviewer 3 who indicated that they addressed the Reviewers' remaining concerns.